# Tunable protein synthesis by transcript isoforms in human cells

Stephen N Floor[1,2]*, Jennifer A Doudna[1,2,3,4,5]*

[1]Department of Molecular and Cell Biology, University of California, Berkeley, Berkeley, United States; [2]Howard Hughes Medical Institute, University of California, Berkeley, Berkeley, United States; [3]Innovative Genomics Initiative, University of California, Berkeley, Berkeley, United States; [4]Physical Biosciences Division, Lawrence Berkeley National Laboratory, Berkeley, United States; [5]Department of Chemistry, University of California, Berkeley, Berkeley, California, United States

**Abstract** Eukaryotic genes generate multiple RNA transcript isoforms though alternative transcription, splicing, and polyadenylation. However, the relationship between human transcript diversity and protein production is complex as each isoform can be translated differently. We fractionated a polysome profile and reconstructed transcript isoforms from each fraction, which we term Transcript Isoforms in Polysomes sequencing (TrIP-seq). Analysis of these data revealed regulatory features that control ribosome occupancy and translational output of each transcript isoform. We extracted a panel of 5′ and 3′ untranslated regions that control protein production from an unrelated gene in cells over a 100-fold range. Select 5′ untranslated regions exert robust translational control between cell lines, while 3′ untranslated regions can confer cell type-specific expression. These results expose the large dynamic range of transcript-isoform-specific translational control, identify isoform-specific sequences that control protein output in human cells, and demonstrate that transcript isoform diversity must be considered when relating RNA and protein levels.

*For correspondence: stephen. floor@gmail.com (SNF); doudna@ berkeley.edu (JAD)

**Competing interests:** The authors declare that no competing interests exist.

## Introduction

Eukaryotic genes can produce a staggering diversity of messenger RNA products. In humans, there is a median of five annotated transcript isoforms per gene, with more than 80 annotated isoforms in some cases (Ensembl release 75). These transcript isoforms arise from the combined action of alternative transcription initiation, splicing and cleavage, and polyadenylation, which are often cell type specific (*Barbosa-Morais et al., 2012*; *Mele et al., 2015*; *Merkin et al., 2012*; *Wang et al., 2008*; *Xiong et al., 2015*; *Zheng and Black, 2013*). As each of these transcript isoforms may in principle harbor a unique set of translational control elements, gene-level expression, which averages out the translational potential of each individual transcript isoform, may not be an accurate measure of protein levels. Since changes in both translation and splicing are linked to numerous human disorders, it is critical to understand the relationship between the two (*Maslon et al., 2014*; *Piccirillo et al., 2014*; *Ruggero, 2013*; *Xiong et al., 2015*).

Eukaryotic mRNAs are decorated with diverse sequence features that can control translation, which can vary between transcript isoforms (*de Klerk and 'tHoen, 2015*; *Ingolia et al., 2011*; *Sterne-Weiler et al., 2013*). Classic examples of translational control include upstream ORFs (uORF)-mediated translational control of yeast *GCN4* (*Hinnebusch, 2005*), protein binding such as the iron regulatory protein (*Gray and Hentze, 1994*), and the action of micro-RNAs (*Nottrott et al., 2006*; *Wilczynska and Bushell, 2015*) or DEAD-box proteins such as eIF4A and Ded1 (*Chuang et al., 1997*; *Hinnebusch and Lorsch, 2012*; *Sen et al., 2015*). Alternative 5′ leader

**eLife digest** To produce a protein, a gene's DNA is first copied to make molecules of messenger RNA (mRNA). The mRNAs pass through a molecular machine known as the ribosome, which translates the genetic code to make a protein. Not all of an mRNA is translated to make a protein; the "untranslated" regions play crucial roles in regulating how much of the protein is produced.

In animals, plants and other eukaryotes, many mRNAs are made up of small pieces that are "spliced" together. During this process, proteins are deposited on the mRNA to mark the splice junctions, which are then cleared when the mRNA is translated. Many different mRNAs can be produced from the same gene by splicing different combinations of RNA pieces. Each of these mRNA "isoforms" can, in principle, contain a unique set of features that control its translation. Hence each mRNA isoform can be translated differently so that different amounts of the corresponding protein product are produced. However, the relationship between the variety of isoforms and the control of translation is complex and not well understood.

To address these questions, Floor and Doudna measured the translation of over 60,000 mRNA isoforms made from almost 14,000 human genes. The experiments show that untranslated regions at the end of the mRNA (known as the 3′ end) strongly influence translation, even if the protein coding regions remain the same. Furthermore, the data showed that mRNAs with more splice junctions are translated better, implying an mRNA has some sort of memory of how many junctions it had even after the protein markers have been cleared.

Next, Floor and Doudna inserted regulatory sequences from differently translated isoforms into an unrelated "reporter" gene. This dramatically changed the amount of protein produced from the reporter gene, in a manner predicted by the earlier experiments. Untranslated regions at the beginning of the mRNAs (known as the 5′ end) controlled the amount of protein produced from the reporter consistently across different types of cells from the body. On the other hand, the 3′ regions can tune the level of protein production in particular types of cells.

Floor and Doudna's findings demonstrate that differences between mRNA isoforms of a gene can have a big effect on the level of protein production. Changes in the types of mRNA made from a gene are often associated with human diseases, and these findings suggest one reason why. Additionally, the ability to engineer translation of an mRNA using the data is likely to aid the development of mRNA-based therapies.

sequences, uORFs, and select tandem 3′ untranslated region (UTR) isoforms have been demonstrated to influence protein production (*Brar et al., 2012*; *Hinnebusch, 2005*; *Ingolia et al., 2011*; *Mayr and Bartel, 2009*; *Sandberg et al., 2008*; *Zhang et al., 2012*). Any of these features may in principle be different between transcript isoforms, but the prevalence and dynamic range of isoform-specific translational control across the human genome is currently unknown.

Previous work measuring genome-wide translation in human cells has focused largely on the relationship between gene-level mRNA abundance and protein levels, which is blind to the contribution of transcript isoforms. Ribosome profiling is not well-suited for measuring transcript isoform-specific translation, primarily due to the short ~30 bp length of ribosome-protected fragments (*Ingolia, 2014*). Prior attempts to characterize isoform-specific translation have measured the effects of 5′ end diversity in yeast (*Arribere and Gilbert, 2013*) and 3′ end diversity in mouse cells (*Spies et al., 2013*), or splicing differences between cytoplasmic and aggregate polysomal mRNAs (*Maslon et al., 2014*; *Sterne-Weiler et al., 2013*). However, sequencing just the ends of transcripts cannot distinguish between transcript isoforms of the same gene harboring degenerate termini. In addition, aggregating polysome fractions averages lowly- and highly-ribosome-associated messages. Therefore, a different strategy is required to understand how the diversity of the human transcriptome impacts translational output.

Here, we adapt a classic approach of polysome profiling coupled with global gene expression analysis (*Arava et al., 2003*) to measure transcript-isoform specific translation using deep sequencing, which we term Transcript Isoforms in Polysomes sequencing (TrIP-seq). By using high gradient

resolution and sequencing depth, this approach yields polysome profiles for over 60,000 individual transcript isoforms representing almost 14,000 protein coding genes. We observe frequent intron retention on ribosome-associated transcripts, even in high-polysome fractions, identifying a population of retained but not nuclear-detained introns (*Boutz et al., 2015*). Properties of 3′ untranslated regions predominate over the 5′ leader sequence as the driving force behind differential polysome association for transcript isoforms of the same gene among the transcript features tested. We show that regulatory sequences differentially included in transcript isoforms of the same gene are modular and can trigger differences in the translation of reporters spanning two orders of magnitude. These findings provide a lens through which to ascribe functional consequences to RNA-seq-generated transcriptomes. Moreover, TrIP-seq analysis uncovers regulatory elements that can be utilized to tune translation of synthetic messages robustly in cells.

## Results

### TrIP-seq measures transcript isoform-specific translation in human cells

We determined the ribosomal association of transcript isoforms by sequencing transcripts cofractionating with different numbers of ribosomes with sufficient depth to determine isoform abundances, as was performed at the gene level in yeast (*Arava et al., 2003*). We treated HEK 293T cells with cycloheximide to stall translation and fractionated the cytoplasm into ribosome-containing samples including one to eight or more ribosomes (*Figures 1A* and *Figure 1—figure supplement 1A*; see Materials and methods for details). We made RNA sequencing libraries from each fraction in biological duplicate and obtained transcript-level abundances using the Cufflinks suite (*Figure 1—source data 1* and *2*; [*Trapnell et al., 2010*]). Clustering of the samples recapitulates the gradient order (*Figure 1B*), indicating the polysome profile was accurately fractionated. Four subgroups emerge from this clustering: the 80S (monosome), low polysomes (two-four ribosomes), high polysomes (five-eight+ ribosomes), and total cytoplasmic RNA. We tested the robustness of the clustering of samples by computing the average Jaccard distance between clusters from data subjected to three different resampling methods, which was ≥0.75, suggesting stable clusters (*Figure 1—figure supplement 1G*; Materials and methods). This suggests that in cells, transcript isoforms are predominantly poorly- or highly-ribosome associated, causing low polysomes to cluster away from high polysomes, which could be have numerous biological origins including highly abundant short ORFs.

To confirm the sequencing data, we first analyzed beta-actin (*ACTB*), which is known to be heavily translated with many ribosomes on each message, and migrates with high polysomes in a gradient (*Figure 1C*; (*Sterne-Weiler et al., 2013*; *Zhang et al., 2015*). However, *ACTB* only has one transcript isoform, so we examined *ATF4*, which has one dominant isoform exhibiting low-polysome association in TrIP-seq data and by RT-PCR (*Figure 1C*), consistent with it being translationally repressed in the absence of cellular stress (*Harding et al., 2000*). We then analyzed eukaryotic elongation factor 1 beta 2 (*EEF1B2*), which has three isoforms with the same coding sequence but alternative 5′ leaders as a representative gene that exhibits isoform-specific ribosome association (*Figure 1—figure supplement 1B*). The isoform-specific polysome profiles and RT-PCR conducted from polysome fractions agree qualitatively, validating the accuracy of the isoform-level quantifications (*Figure 1D*). Weaker amplification of the longer isoform (EEF1B2-003) in high polysomes may be due to PCR bias towards smaller amplicons (*Walsh et al., 1992*), and replicate experiments show EEF1B2-003 present in high polysomes (*Figure 1—figure supplement 1C*). We additionally validated the TrIP-seq data using qRT-PCR from polysome fractions and find that the two measurements agree for both *EEF1B2* and *SRSF5* (*Figure 1—figure supplement 2*). Both coding mRNAs and long noncoding RNAs (lncRNAs) associate with ribosomes (*Ingolia et al., 2014*; *van Heesch et al., 2014*), so we measured their polysomal abundance. Protein-coding genes and isoforms are found across the polysome, but noncoding and lncRNA genes are predominantly in the low polysome fractions indicating that they are generally weakly ribosome associated or in other large macromolecular complexes (*Figure 1E* and *Figure 1—figure supplement 1D*).

To determine whether TrIP-seq is a measure of translation as opposed to cryptic association with large macromolecular complexes or stalled ribosomes (*Darnell et al., 2011*; *Ishimura et al., 2014*), we compared TrIP-seq data to ribosome profiling and proteomics datasets. First, we reanalyzed a ribosome profiling dataset from HEK 293T cells and compared the number of ribosome-protected

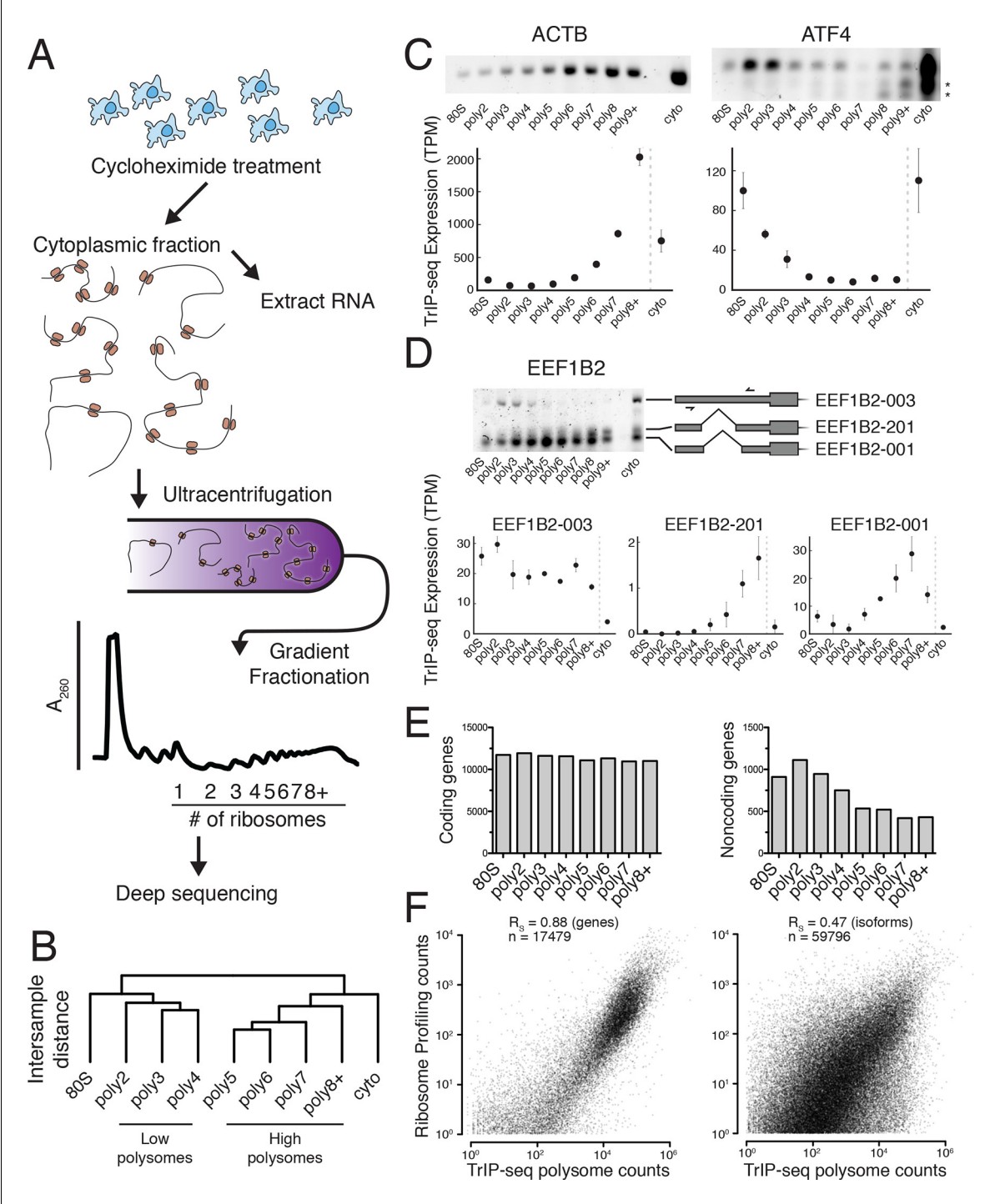

**Figure 1.** Transcript Isoforms in Polysomes sequencing (TrIP-seq) measures transcript isoform specific translation. (A) HEK 293T cells were treated with cycloheximide and the cytoplasmic fraction was extracted and applied to a sucrose gradient, which was further fractionated into individual polysomes that were converted into sequencing libraries. (B) Intersample clustering recapitulates the gradient order of polysomes, indicating the sequenced fractions are faithful to the gradient profile. (C,D) RT-PCR analysis and transcript-level quantification for *ACTB* and *ATF4* (C) and three transcripts of *EEF1B2* (D) demonstrating concordance of sequencing and transcript-specific RT-PCR. TPM – transcripts per million. *nonspecific amplicon. (E) Members of transcript classes with more than 100 reads in indicated fractions show that coding genes are represented across the polysome while noncoding genes are preferentially in low polysome fractions. (F) Spearman's correlation ($R_S$) between gene (left) or isoform (right) read counts from ribosome profiling or TrIP-seq. See also *Figure 1—figure supplement 1* and Materials and methods for calculation of TrIP-seq polysome counts.

*Figure 1 continued on next page*

*Figure 1 continued*

The following source data and figure supplements are available for figure 1:

**Source data 1.** Gene-level abundances for all Ensembl 75 annotated human genes across all sequenced polysome fractions.
**Source data 2.** Transcript isoform abundances for all Ensembl 75 annotated human transcripts across all sequenced polysome fractions.
**Figure supplement 1.** Extended TrIP-seq validation.
**Figure supplement 2.** Extended validation of TrIP-seq isoform abundances across polysome fractions using qRT-PCR.
**Figure supplement 3.** Read Tracking Across the Sequencing Analysis Pipeline.

fragments to a weighted sum of the TrIP-seq polysome reads (Materials and methods; $n_{ribo} \geq 2$) (*Sidrauski et al., 2015*). We observe a strong correlation between ribosome profiling and TrIP-seq at the gene level ($R_S = 0.88$; *Figure 1F*). However, at the isoform level, the correlation decreases ($R_S = 0.47$), which is worse than the correlation between TrIP-seq replicates ($R_S = 0.90$; *Figure 1—figure supplement 1E*), suggesting the discrepancy is not due to variability in TrIP-seq but instead is likely due to known issues quantifying transcript isoforms in ribosome profiling (*Ingolia, 2014*). We next computed the translation efficiency (TE) of ribosome profiling and TrIP-seq data by dividing by cyto-plasmic RNA levels. TE measured by the two methods correlated with $R_S = 0.24$, perhaps due to var-iability in culture conditions or in either technique. We compared the data of Sidrauski *et al* to an unpublished ribosome profiling dataset in 293T cells from the Yoon-Jae Cho lab. Ribosome-pro-tected fragments between experiments correlate with $R_S = 0.78$, while TE has a substantially lower correlation of $R_S = 0.41$, suggesting biological or technical variability disproportionately affects TE. We then compared TrIP-seq data to protein abundances from HEK 293T cells, and find that the two datasets correlate ($R_S = 0.57$, *Figure 1—figure supplement 1F*, (*Geiger et al., 2012*), which is bet-ter than RNA-seq ($R_S = 0.45$; not shown). The observations that most translational stalling in mam-malian cells is transient and translation elongation rates are homogeneous (*Ingolia et al., 2011*), TrIP-seq appears to underestimate rather than overestimate protein abundance (*Figure 1—figure supplement 1F*), and TrIP-seq correlates well at the gene-level with ribosome profiling (*Figure 1F*) suggest that TrIP-seq primarily measures translating ribosome association as opposed RNAs bound to other large complexes.

## Diverse human isoform-specific polysome association patterns

To extract global trends in isoform-specific translation, we hierarchically clustered transcript isoform polysome profiles and selected eight clusters that are representative of general trends in the data (*Figure 2*; Materials and methods). The depth of sequencing (*Figure 1—figure supplement 3*), aug-mented by the fractionation strategy, enables detection of 62,703 transcript isoforms in the poly-some profile (*Figure 2—source data 1*). Isoforms in the observed clusters exhibit diverse average patterns across polysomes (*Figure 2A,B*), from clusters 1 and 2, which contain isoforms primarily in high polysomes, to cluster 3 with isoforms in the middle, to clusters 6 and 7 where isoforms are in low polysomes. Independent clustering of the two biological replicate TrIP-seq datasets shows that the high- and low-polysome clusters (1, 2 and 6) appear more robust when comparing between aver-aged and individual replicate clusterings (*Figure 2—figure supplement 1A*). Many clusters have sim-ilar total polysome abundance but different distributions, indicating that to obtain accurate measurements of isoform-specific translatability it is crucial to fractionate the polysome profile.

Surprisingly, the poorly translated cluster 7, which has ~9,000 transcripts, contains a similar num-ber of annotated retained intron or protein-coding isoforms (*Figure 2C*). We queried the gene ontology (GO) terms associated with each cluster and found that the poorly translated clusters 6 and 7 are enriched for translation and splicing genes (*Figure 2—figure supplement 1B*), implicating the alternative splicing-nonsense-mediated decay (AS-NMD) pathway (*Jangi et al., 2014*; *Lareau et al., 2007*). However, not all retained intron transcripts are subject to NMD (*Boutz et al., 2015*; *Gohring et al., 2014*), and we also observe retained intron transcripts in the best-translated clusters (1 and 2), indicating at least some of the introns may be translated. In sum, clustering of isoform

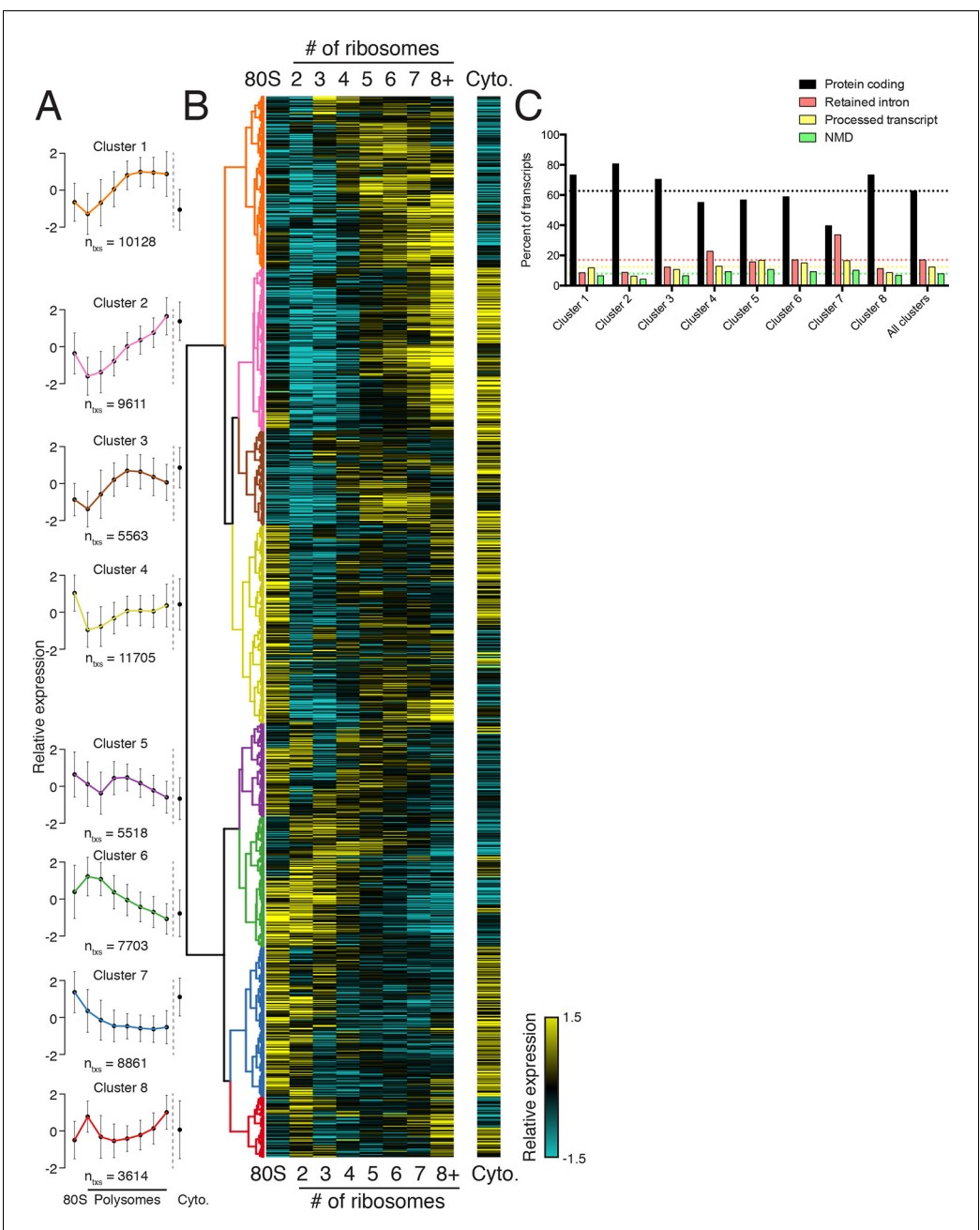

**Figure 2.** Clustering of transcript distributions yields eight major clusters with diverse behavior across the polysome profile. (A) The average relative abundance of all isoforms in each cluster across polysomes is shown. Error is s.d. (B) Hierarchical clustering of 62,703 transcript isoform distributions across the polysome profile and cytoplasmic fraction. Yellow: above isoform average, cyan: below isoform average. (C) Transcript type distribution per cluster from Ensembl-annotated biotypes. Dotted lines mark the abundance of each transcript type in all isoforms that went in to the clustering. See also *Figure 1—source data 1* and *2* and *Figure 2—figure supplement 1*.

The following source data and figure supplement are available for figure 2:

**Source data 1.** Variance stabilized transcript isoform abundances (see Materials and methods) and cluster number for all Ensembl 75 annotated human transcripts across all sequenced polysome fractions.

**Figure supplement 1.** Further information on TrIP-seq clusters.

abundance distributions across polysomes immediately provides insight into the diverse patterns of ribosome association by individual isoforms and transcript types in cells, and reveals intron-rich clusters of transcripts associated with polysomes that escape nuclear detention.

## 3′ UTRs and introns drive isoform-specific polysome association

We reasoned we could extract features regulating the translation of a gene by comparing transcript isoforms of the same gene that are well- or poorly-translated. We selected 24 features to explore that are involved in translational control, such as the length of the coding sequence and untranslated regions, predicted secondary structure, and microRNA binding sites. As their average polysome profiles and composition are broadly similar (*Figure 2*), we merged clusters 1 and 2 to generate a larger pool of high-polysome isoforms (*Figure 3A*), and compared these to cluster 6, representing poorly translated isoforms in low polysomes. We then compared transcript isoforms of the same gene that are in high- and low-polysome clusters, or gene-linked isoforms, to extract transcript features that influence translation. The number of gene-linked isoforms per feature per set varies between 569 and 6491. We then measured the effect size and calculated statistical significance for all features between gene-linked isoforms (*Figures 3B* and *Figure 3—figure supplement 1A*; Materials and methods; [*Cliff, 1993*]).

We find that longer coding sequences and highly abundant versions of gene-linked isoforms are biased towards high polysomes, likely because shorter coding sequences cannot accommodate as many ribosomes (*Figure 3B* and *Figure 3—figure supplement 1A*). However, the length of the coding sequence is not the sole determinant of polysome association, since the ribosome density (measured by dividing the weighted sum of TrIP-seq polysome reads by the number of cytoplasmic reads) is also higher in gene-linked isoforms found in high polysomes (*Figure 3B*). We also find that transcripts from gene-linked isoforms with more exons also tend to be better translated, as was observed using reporter genes (*Nott et al., 2003*; *2004*). This is not due to overall transcript length, as there is no significant difference between transcript length in gene-linked isoforms, reflecting a positive influence of splicing on translation. Highly translated isoforms contain fewer rare codons on average, but contain more stretches of rare codons, perhaps indicating a need for translational pausing (*Figure 3B*). The observation that exons promote translatability has been shown for select genes, and here we show that this extends across the human genome.

One of the strongest effects seen on polysome association of gene-linked isoforms comes from the length and content of the 3′ UTR (*Figure 3B* and *Figure 3—figure supplement 1A*). Specifically, we find that gene-linked 3′ UTRs in low polysomes are considerably longer than those on high polysomes (mean length 1551 nt versus 982 nt). There are numerous regulatory elements contained within 3′ UTRs, including microRNA binding sites, AU-rich elements, and protein-binding sites (*Bartel, 2009*; *Szostak and Gebauer, 2013*). The fraction of the 3′ UTR containing AU-rich elements is increased in low polysome gene-linked isoforms, possibly due to translational repression (*Brooks and Blackshear, 2013*; *Moore et al., 2014*). Conserved predicted binding sites for miRNAs (*Garcia et al., 2011*) are also more abundant in poorly translated gene-linked isoforms, but this could be due to a correlation between 3′ UTR length and the number of miRNA binding sites. We, therefore, filtered the miRNAs to those bound to AGO1 or AGO2 in HEK 293T cells (*Ender et al., 2008*). In this HEK 293T-expressed miRNA set, the difference for both the number of conserved miRNA binding sites and their score increases between gene-linked isoforms, suggesting that miRNA binding may be functionally relevant in this context (*Figure 3B* and *Figure 3—figure supplement 1A*). Increased AU-rich elements and miRNA binding should lead to decreased half-life for low polysome isoforms, which is borne out in half-life comparisons (*Figure 3B* and *Figure 3—figure supplement 1A*; [*Tani et al., 2012*]).

Surprisingly, gene-linked isoforms in high polysomes have increased predicted structure both at the cap and in 75-nucleotide windows in the 5′ leader. There are at least three possibilities that could explain this result. First, RNA structures can both repress or promote translation in a context dependent manner (*Xue et al., 2015*), so it is possible that some isoforms are being driven to high polysomes by recruitment of transacting factors. Second, recent studies of global RNA secondary structure have consistently observed different structure in cells than in vitro (*Rouskin et al., 2014*; *Spitale et al., 2015*). Lastly, HEK 293T cells may translate messages with inhibitory 5′ leaders more efficiently than other cell types. In contrast, we find no significant dependence for the length of the

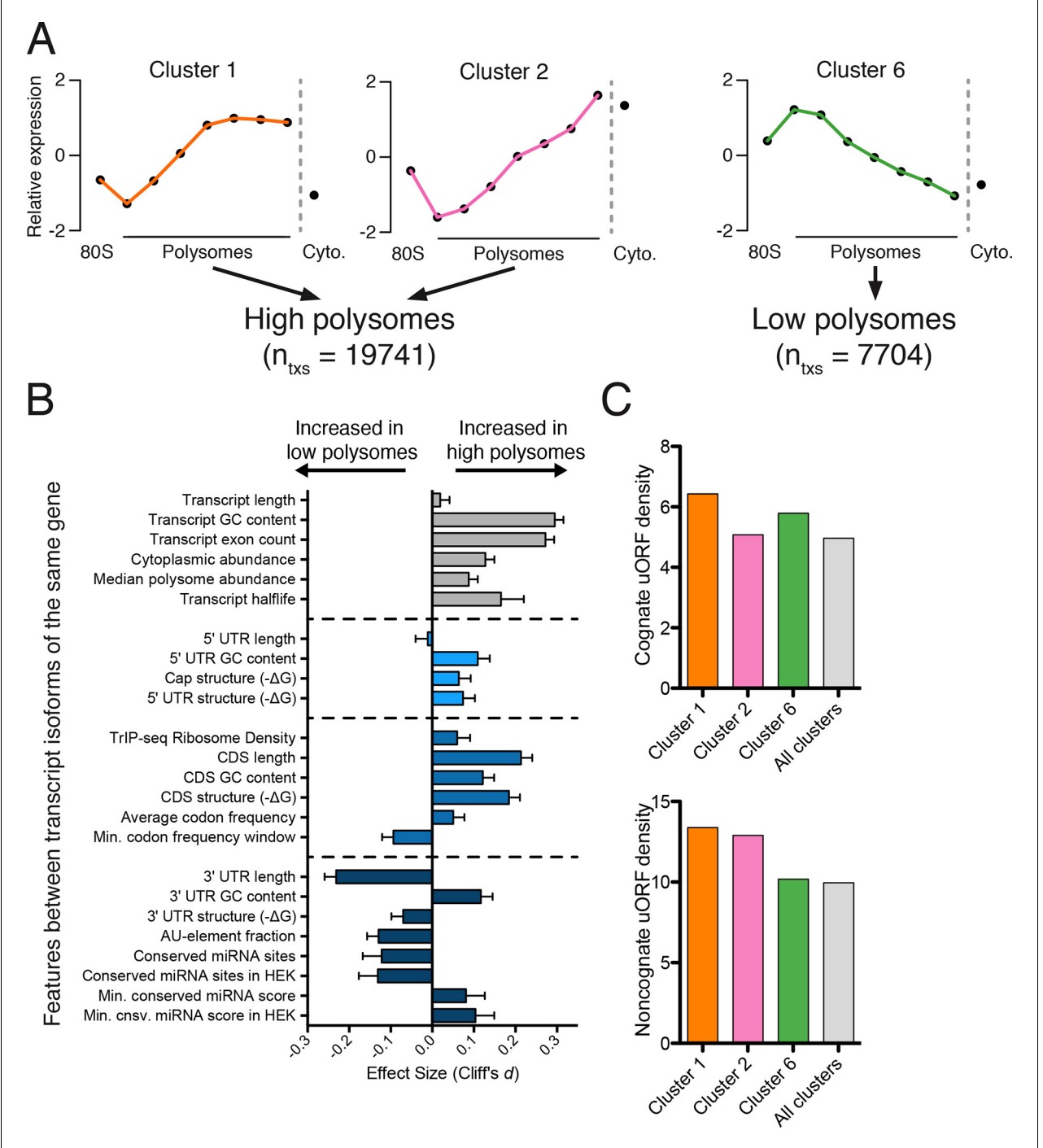

**Figure 3.** Effect of transcript features on polysome association. (A) Meta-transcript distributions for two high polysome clusters (1 and 2) and one low polysome cluster (6). Clusters 1 and 2 were pooled for the analysis in (B). (B) The distance between distributions for 24 different transcript features evaluated for transcripts strongly or weakly associated with polysomes. Distance is the nonparametric effect size, measured as the dimensionless quantity Cliff's *d* (see Materials and methods) and error bars are bootstrapped 95% confidence intervals. All differences except 5' UTR and transcript length are significant at the p = 0.001 level based on two-tailed Mann-Whitney U-tests (*Figure 3—figure supplement 1A*). See Materials and methods for a description of all features and how they were tabulated. UTR – untranslated region; CDS – coding sequence. (C) Enrichment of either cognate (ATG start codon) or non-cognate (non-ATG) uORFs in high polysome versus low polysome clusters. Density is uORFs per 100 isoforms. See also *Figure 3—figure supplement 1*.

The following figure supplement is available for figure 3:

*Figure 3 continued on next page*

*Figure 3 continued*

**Figure supplement 1.** Further details of high versus low polysome associated transcript isoform comparisons.

5′ leader region on polysome association based on this comparison (*Figures 3B* and *Figure 3—figure supplement 1A*). However, isoforms containing 5′ leaders over 1000 nucleotides long are poorly ribosome-associated relative to shorter 5′ leaders (*Figure 3—figure supplement 1B*) and cluster 7, which is associated with few ribosomes, contains longer 5′ leaders (*Figure 3—figure supplement 1C*). It is likely that more subtle features of the 5′ leader also influence translatability, as in the examples in *Figure 1D*.

Upstream open reading frames (uORFs) can positively or negatively influence translation (*Brar et al., 2012*; *Calvo et al., 2009*; *Ferreira et al., 2013*; *Hinnebusch, 2005*). We, therefore, counted the number of experimentally determined uORFs (*Wan and Qian, 2014*) in each cluster and observe surprisingly complex behavior. Cognate uORFs (those starting with ATG) promote or repress translation, while noncognate uORFs generally promote translation (*Arribere and Gilbert, 2013*; *Brar et al., 2012*). Along these lines, we find that cluster 1 (high polysomes) is enriched for cognate and noncognate uORFs while cluster 6 (low polysomes) is enriched for cognate uORFs. However, cluster 2 (high polysomes) is enriched for only noncognate uORFs, indicative of the complex and idiosyncratic behavior of uORFs. Taken together, we find predominant influences for the 3′ UTR and the number of introns in determining polysome occupancy of gene-linked transcript isoforms in human cells, while a diversity of other features can influence translatability to a lesser extent.

## Predicting translation changes during preimplantation human development

RNA-seq characterizes the isoform diversity of a sample, but not the functional consequences of this diversity. We reasoned that TrIP-seq data could be used to predict which isoform changes are likely to lead to translation changes in other systems. We chose a human embryonic dataset containing 124 individual cells from seven preimplantation developmental stages and reprocessed these data to directly compare to TrIP-seq data (Materials and methods; [*Yan et al., 2013*]). We mapped transcripts expressed during each developmental stage onto the TrIP-seq clusters, to attempt to gain insight into global translational properties in human embryos (*Figure 4A*). In embryos, our analysis predicts a shift towards the extremes of translation, with an increase in the percent of transcript isoforms that are both highly (e.g. clusters 1 and 2) and lowly (e.g. cluster 6) translated isoforms in HEK 293T cells (*Figure 4A*). Localized translation is widespread in development, and it is possible that the observed increase in poorly translated isoforms reflects a greater need for translational control (*Besse and Ephrussi, 2008*; *Jung et al., 2014*). It is also possible that translation in early embryos is differentially regulated than in HEK 293T cells, which is now testable by applying TrIP-seq to other cell types, yielding cell type-specific translational control programs.

We then collected genes with isoforms that change between early and late embryonic stages. Clustering of abundances for 45,895 isoforms across embryogenesis yielded seven clusters with varying profiles, which we then analyzed by comparing transcripts of the same gene located in different embryonic clusters as before (*Figure 4—figure supplement 1A*). These isoforms were filtered by those that move between low- and high-polysome clusters in the TrIP-seq data, yielding 366 isoform pairs belonging to 270 genes that are developmentally regulated and exhibit differential translation in HEK 293T cells. For example, the *CSDE1* gene expresses one isoform in oocytes, early development and in human ES cells, which is poorly translated (CSDE1-002), but a second appears following fertilization that is well-translated (CSDE1-007; *Figure 4B,C*). As global zygotic transcription begins in the two-cell stage (*Vassena et al., 2011*; *Yan et al., 2013*), it is possible that the alternative *CSDE1* isoform is supplied by the spermatid (*Fischer et al., 2012*; *Soumillon et al., 2013*) or that it may be precociously transcribed. We also present three other genes (*EIF4A2, RNF170,* and *TBC1D15*) that show stage-specific expression in the embryo data and differential polysome association in HEK 293T cells (*Figure 4—figure supplement 2*). Therefore, widespread changes at the

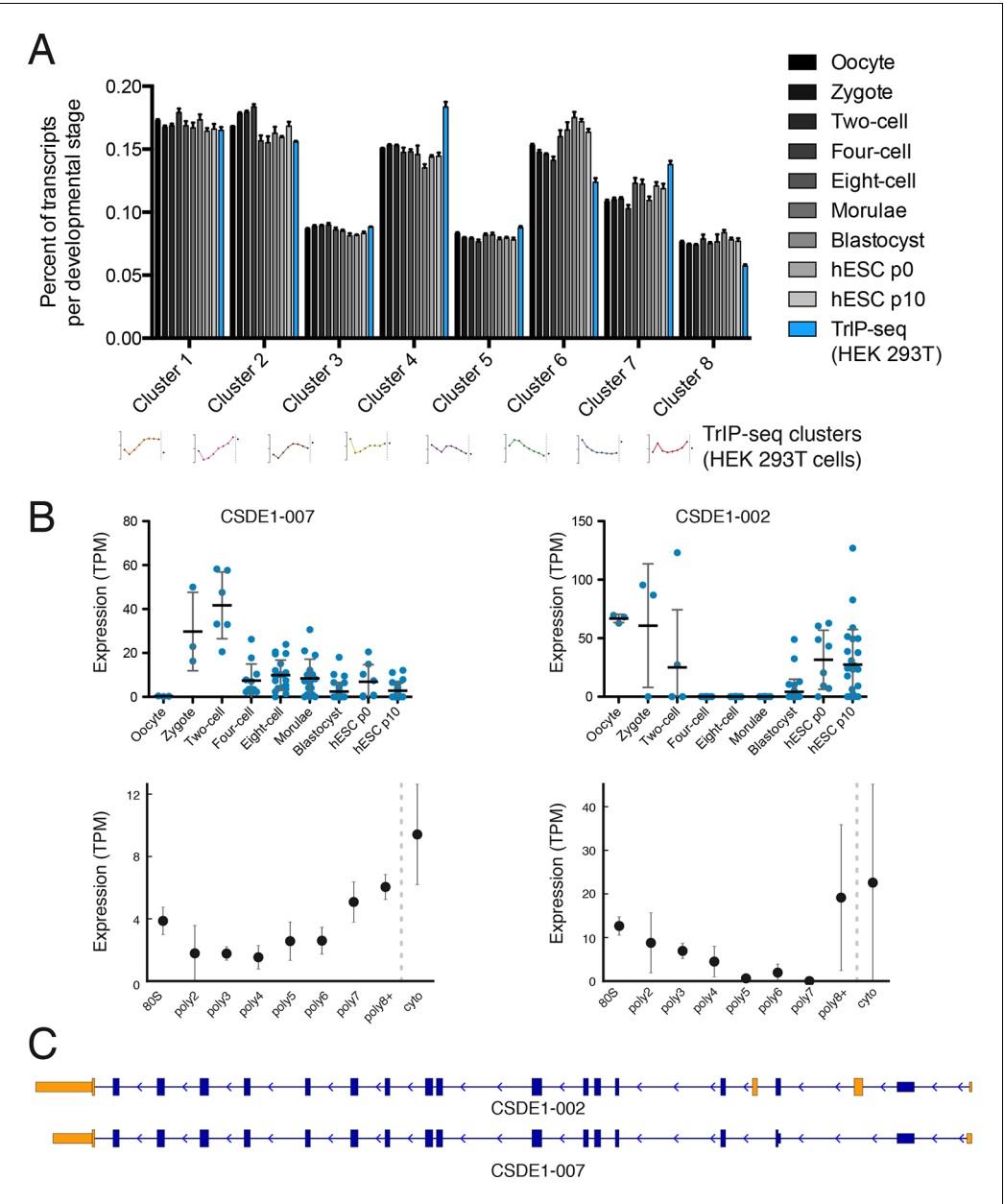

**Figure 4.** Predicted isoform-specific translational control changes during human embryogenesis. (**A**) For each embryonic stage, transcripts are mapped onto all eight TrIP-seq clusters, as in *Figure 2*. The percentage of transcripts mapping to each cluster per stage is then calculated, and compared to the percentage for TrIP-seq data. hESC – human embryonic stem cell; p0, p10 – passage zero or passage ten. Error is s.d. between single cells at each embryonic stage or between TrIP-seq biological replicates. (**B**) Expression levels for two transcripts of *CSDE1* across embryonic development and polysome fractions demonstrating a switch in translational status. Error is S.D. TPM – transcripts per million. (**C**) Diagram of two transcript isoforms of CSDE1. Regions different between the two isoforms are in yellow and select shared intronic regions have been shortened for clarity. See also *Figure 4—figure supplement 1*.

The following figure supplements are available for figure 4:

**Figure supplement 1.** Clustering of human preimplantation embryo data

**Figure supplement 2.** Additional examples of transcript isoforms that exhibit differential expression in human embryos and differential translation in TrIP-seq data.

transcriptome level during human embryogenesis may produce concomitant changes in protein production, which can now be predicted using TrIP-seq data.

## Tunable translation by isoform-specific regulatory elements

Engineering the translation of a transcript without altering its coding sequence is desirable when introducing exogenous mRNA to cells or patients. We hypothesized that the regulatory features we discovered by comparing highly to lowly translated isoforms (*Figure 3B*) should be transferrable to an arbitrary reporter. Nine different 5′ leaders and eight different 3′ UTRs derived from gene-linked isoforms (*Figure 3*) were appended onto *Renilla* luciferase with a synthetic poly-A$_{60}$ tail, which were individually in vitro transcribed, capped, and 2′-O-methylated (*Figure 5—figure supplement 1A* and Materials and methods). We elected to transcribe RNA both because plasmid-encoded transcripts can be heterogeneous and to mimic a scenario where one is delivering RNA to affect cell activity or for therapeutic intervention (*Kormann et al., 2011*; *Warren et al., 2010*).

All tested 5′ leader sequences modulate protein production by the luciferase reporter in concordance with the observed polysome association by TrIP-seq, when transfected into HEK 293T cells (*Figure 5A,B*). The three 5′ leaders from *EEF1B2* alter luciferase production in a stepwise manner by roughly a factor of 20. The TrIP-seq profiles from *NAE1* are the most distinct among those tested, and the two luciferase constructs differ in output by two orders of magnitude. Surprisingly, the two 5′ leaders from *RICTOR* differ by only nine nucleotides, yet still exhibit differential protein production, possibly due to altered local RNA secondary structure near the 5′ cap (*Figure 5—figure supplement 1B*). Both 5′ leaders from *SRSF5* contain two uORFs, but they are close (SRSF5-002) or far (SRSF5-005) from the start codon (*Figure 5—figure supplement 1A*), suggesting reinitiation following uORF translation may be impacting luciferase production (*Grant et al., 1994*; *Hinnebusch, 2005*). Not only do these data provide strong evidence that TrIP-seq data can be used to predictably tune the output of heterologous mRNAs using isoform-specific untranslated regions, it additionally validates that the polysome profiles observed are connected to translational output.

We additionally selected eight 3′ UTRs from four sets of gene-linked isoforms to test their ability to control translation, and find good agreement with the TrIP-seq polysome abundance for two out of four 3′ UTR pairs (*Figure 5C,D*). In all cases, replacing the 3′ UTRs leads to decreased protein production compared to the short 3′ UTR in the control RNA. The paired 3′ UTRs of *NAB1* and *RICTOR* each differ by two kilobases, and a factor of 35 and 44 in luciferase output, respectively, indicating that 3′ UTRs can strongly modulate protein production (*Mayr and Bartel, 2009*; *Sandberg et al., 2008*). However, the two other paired 3′ UTRs, despite also differing by two kilobases each, are not distinguishable at the protein production level in cells. The two tested *CCNE2* isoforms differ in both the 5′ and 3′ UTRs, so it is possible that the isoform-level translational control is occurring via the 5′ UTR, and there is a small but not significant difference between the two *NDC1* 3′ UTRs, suggesting regulation of this 3′ UTR may be subtle. We also find a positive relationship between the average number of ribosomes on each transcript isoform in TrIP-seq and the luciferase fold change (*Figure 5E*). In sum, we show that elements found to control translation at the isoform level using TrIP-seq can be grafted on to heterologous coding sequences to control translation over a range of two orders of magnitude in human cells.

## Translational control of reporters by the 5′ leader is robust between cell types

As different cell types contain different macromolecules, they may also translate messages with the same regulatory features differently. We, therefore, tested the panel of gene-specific untranslated regions fused with luciferase (*Figure 5*) in four additional cell lines: A549 (lung carcinoma), K-562 (chronic myelogenous leukemia), MCF-7 (breast adenocarcinoma), and Hep G2 (hepatocellular carcinoma). We additionally tested HEK 293T cells at a shorter timepoint of 2 hr, to explore the role of RNA stability in the observed luciferase output.

Remarkably, the luciferase production from all tested 5′ leader reporters was qualitatively similar in all six conditions (*Figure 6A*). However, the difference between the two *RICTOR* 5′ leader sequences is decreased. In contrast, the 3′ UTR reporters show considerably more variability (*Figure 6B*). Calculation of the coefficient of variation of all reporters across all six conditions highlights the greater variability found in 3′ UTR reporters (*Figure 6—figure supplement 1A*). We additionally

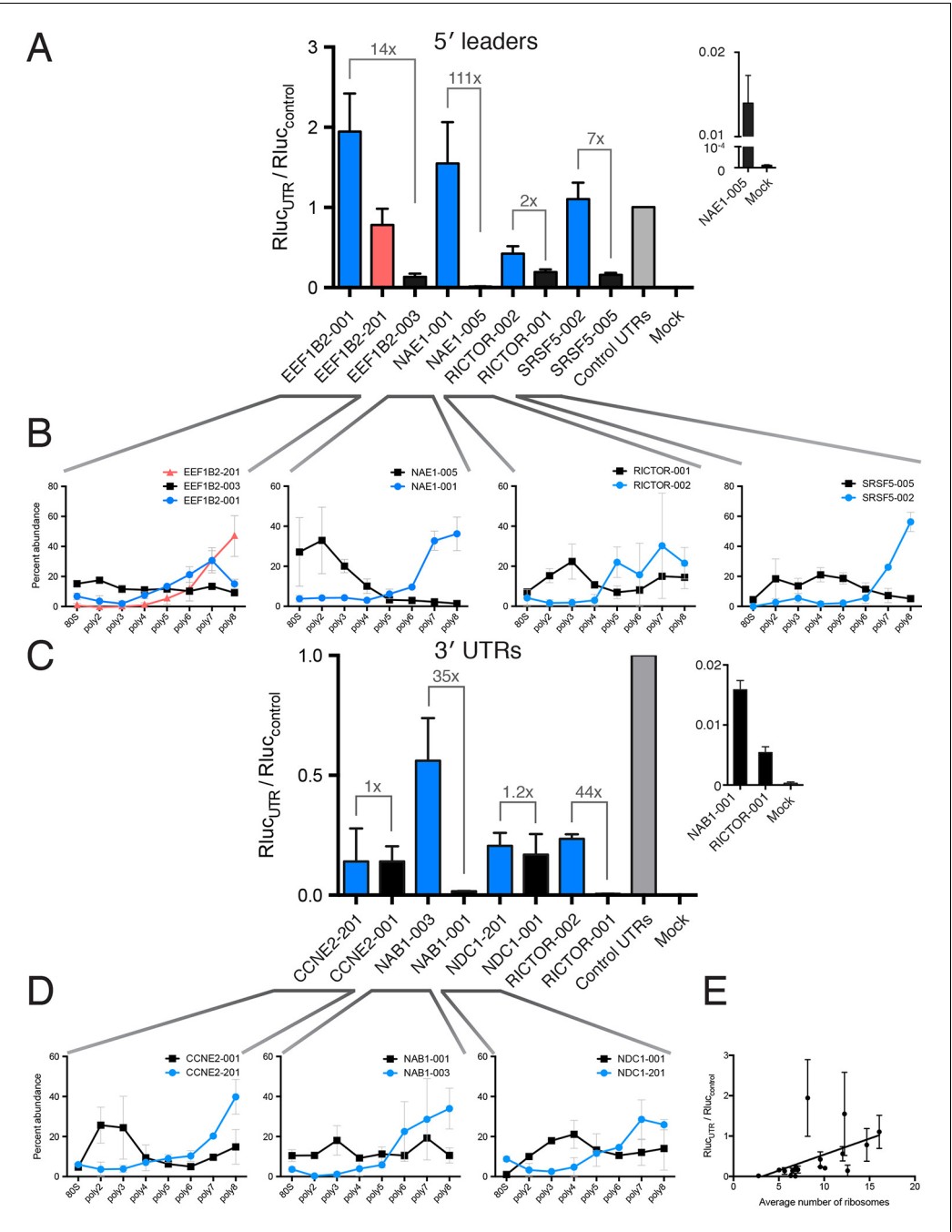

**Figure 5.** Regulatory UTR sequences are sufficient to control translation of a heterologous reporter. HEK 293T cells were transfected with indicated RNAs and *Renilla* luciferase units are plotted. (**A–D**) *Renilla* light units normalized to control UTR mRNA (**A,C**) and corresponding traces from TrIP-seq data (**B,D**) for 5′ leaders (**A,B**) or 3′ UTRs (**C,D**) exhibit differential protein production over two log-units in cells. Inset graphs show comparisons between mock-transfected cells and the lowest output mRNAs. Error bars are S.E.M. from at least four biological and three technical replicates (twelve total; A) or three biological and three technical replicates (nine total; B). (**B, D**) Error is S.D. Rluc – *Renilla* luciferase units; UTR – untranslated region. (**E**) Luciferase fold change versus the average number of ribosomes on each transcript, computed by averaging the plots in B and D. See also *Figure 5—figure supplement 1*.

The following figure supplement is available for figure 5:

**Figure supplement 1.** Diagrams of isoforms differentially represented in gigh versus low polysomes.

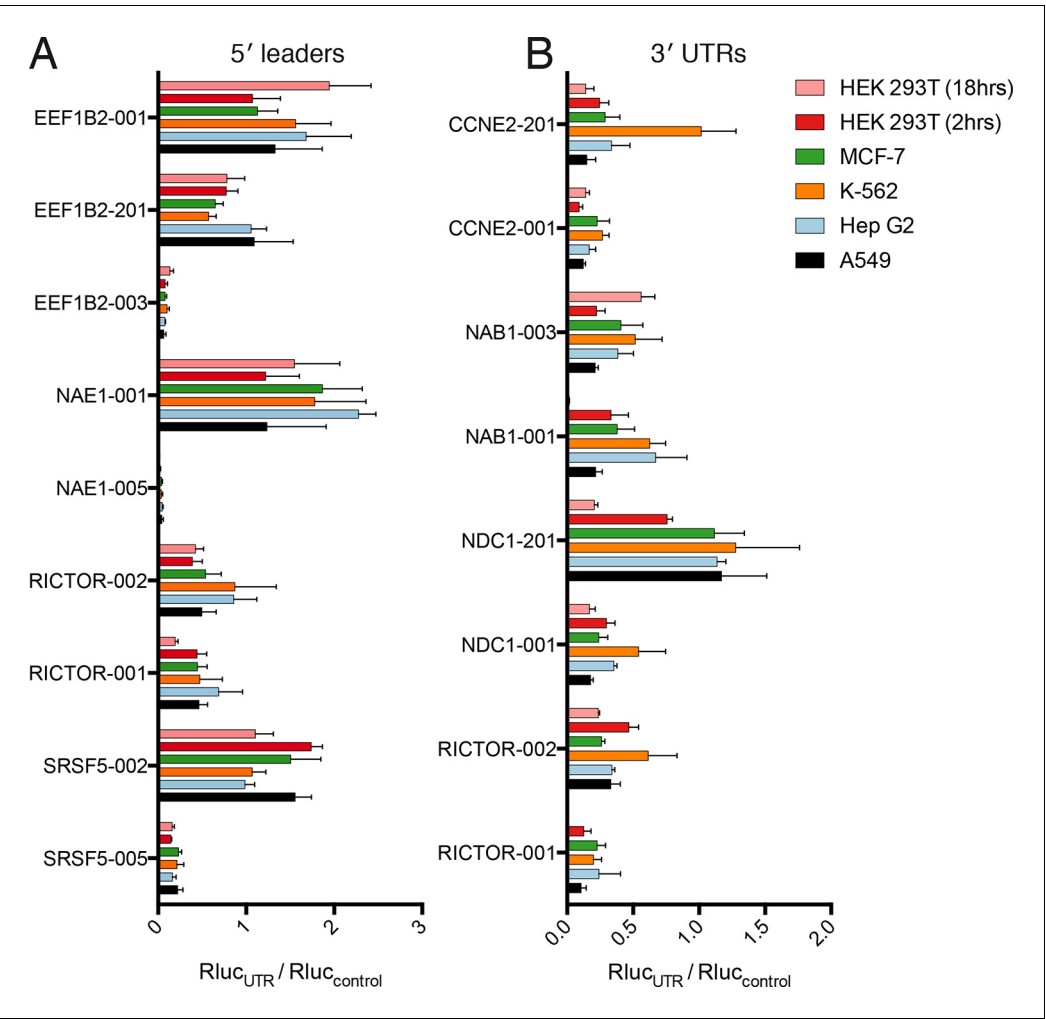

**Figure 6.** Translational control by transcript 5′ leader sequences is robust across cell types. (**A,B**) Five cell lines in six conditions were transfected with *Renilla* luciferase fused to the UTR indicated and luciferase units normalized to control UTR mRNA. Both 5′ leaders (**A**) and 3′ UTRs (**B**) were tested. Error is S.E.M. between three technical and three biological replicates (nine total) per condition. See also *Figure 6—figure supplement 1*.

The following figure supplement is available for figure 6:

**Figure supplement 1.** Changes in protein production conferred by 5′ UTRs are more robust across cell lines than those conferred by 3′ UTRs.

compared HEK 293T cells at 2-hr and 8-hr post-transfection to ascertain differences in RNA stability. Most 5′ leader reporters are similar between these two timepoints (*Figure 6A*), but 3′ UTR reporters show larger changes. Specifically, NAB1-001 and RICTOR-001 generate much more luciferase at the 2-hr timepoint, suggesting the long UTRs of these genes may promote RNA degradation leading to decreased protein at eighteen hours.

Thus, translational control by the 5′ leader sequence is robust to changes in macromolecular composition across the cell lines tested, perhaps because the abundance of most translation initiation factors is high (*Kulak et al., 2014*), which will buffer against small changes in their expression level. However, factors interacting with 3′ UTRs, specifically miRNAs, can vary considerably between cell types, and this variability may be the cause of the diverse behavior of the 3′ UTR reporters across cell lines. Therefore, at least for the panel tested here, transcript 5′ leaders confer robust control of

protein production across cell types, while transcript 3′ UTRs may be better suited to tune the production of a protein to a particular cell type.

## Discussion

In this work, we showed that the dynamic range of transcript-isoform-specific translational control spans at least two orders of magnitude in human cells (*Figure 5*), indicating that it is crucial to take isoform-level effects into account when assessing translation in organisms with extensive alternative transcript processing (*Figure 7A*). Globally, we find that alternative 3′ UTRs broadly influence translation, with longer isoforms of the same gene associated with lower protein production (*Figure 3B*). We further show that regulatory regions are sufficient to control the translation of unrelated coding sequences, enabling predictable tuning of translational output of arbitrary genes (*Figures 5* and

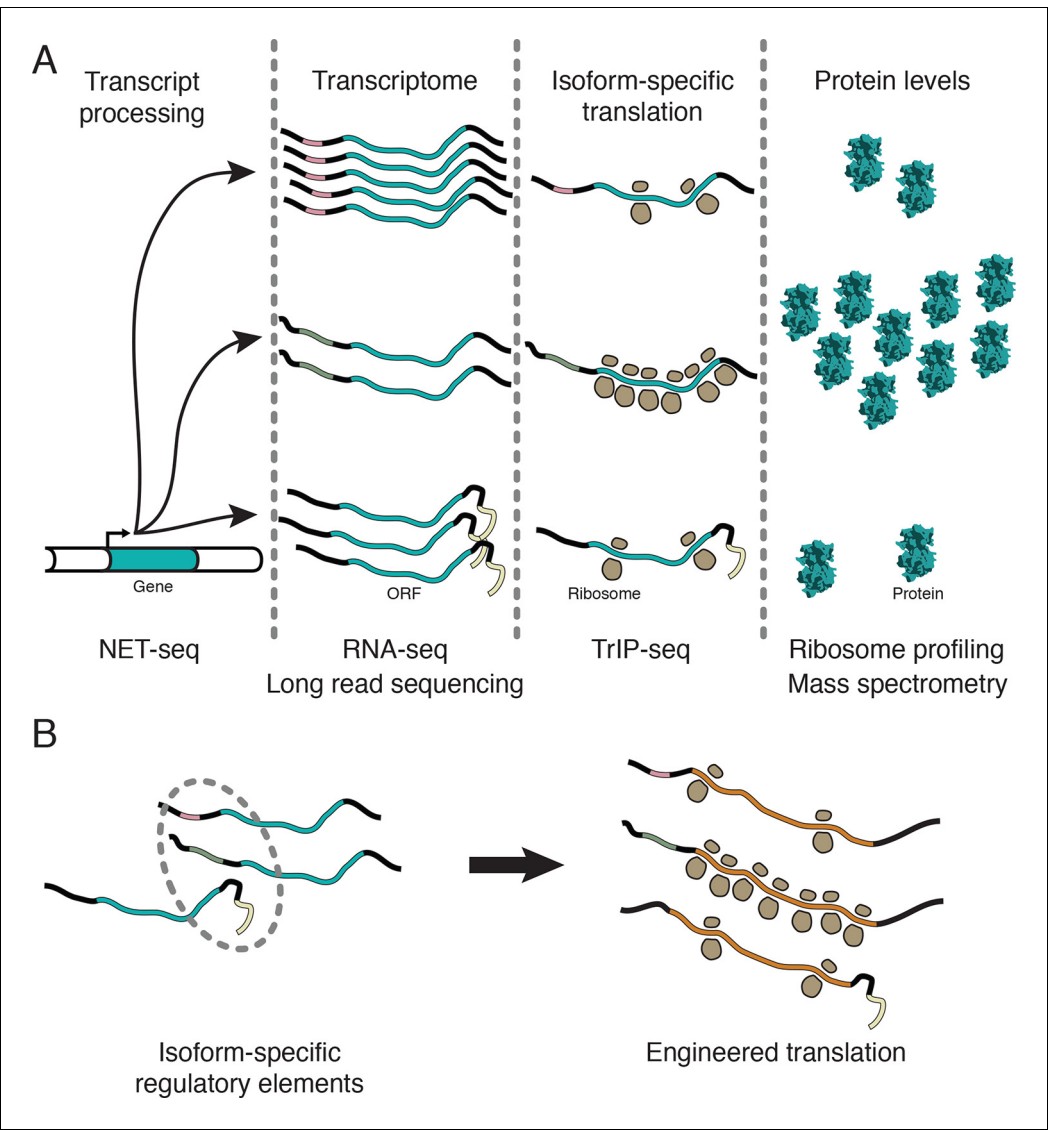

**Figure 7.** Transcript processing has widespread effects on protein production. (**A**) The mapping between gene and protein is complex, and RNA processing has a strong influence on the translatability of individual transcript isoforms and therefore protein production. (**B**) Regulatory elements can be discovered by comparing transcript isoforms that are differentially translated, and subsequently fused to heterologous genes to control protein output in cells. ORF – open reading frame.

6B). We demonstrate that translational control conferred by a panel of 5′ leaders is more robust across cell types than 3′ UTRs, suggesting predictable control of protein production. Future work focused on measuring isoform-specific translation in different cell types will yield cell type-specific regulatory sequences, which could then be used to engineer cell type-specific translation of mRNA, as in derivation of pluripotent stem cells (*Warren et al., 2010*), or mRNA therapeutics (*Kormann et al., 2011*).

Several observations indicate that TrIP-seq is a faithful measure of translation in cells. First, TrIP-seq and ribosome profiling are highly correlated at the gene level (*Figure 1F*) and both ribosome profiling and TrIP-seq correlate with mass spectrometric measurements of protein abundance ([*Ingolia et al., 2009*], *Figure 1—figure supplement 1F*). Second, protein-coding genes are enriched in high ribosomal fractions (*Figure 1E*), while lncRNAs are found primarily in the low ribosomal fractions (*Figure 1E* and *Figure 1—figure supplement 1D*). Third, a systematic investigation of translational stalling in mouse embryonic stem cells found this phenomenon to primarily be transient and not lead to ribosome accumulation on transcripts (*Ingolia et al., 2011*), although in specific circumstances it can be more widespread (*Darnell et al., 2011*; *Ishimura et al., 2014*; *Richter and Coller, 2015*). Fourth, known highly- and lowly-translated transcript isoforms are enriched in high- and low-polysome fractions in TrIP-seq data (*Figure 1C*). Fifth, reads derived from stalled polysomes would artificially inflate the apparent translatability, however, TrIP-seq primarily underestimates rather than overestimates protein abundance (*Figure 1—figure supplement 1F*). Sixth, cryptic 'pseudo-polysomes' induced by miRNA complexes would preferentially enhance polysome association for long 3′ UTR transcript isoforms (*Maroney et al., 2006*; *Nottrott et al., 2006*; *Olsen and Ambros, 1999*; *Thermann and Hentze, 2007*), which is the opposite of what is observed (*Figure 3B*). Lastly, transcript isoform changes found to lead to differential ribosome association using TrIP-seq are sufficient to modulate the protein output from a reporter RNA in a manner predicted by the TrIP-seq data (*Figure 5*), indicating that transcript-specific ribosome association is a correlate of protein output. Therefore, we conclude that TrIP-seq is a measure of transcript-specific translation.

What are the mechanistic foundations of the observed transcript-specific translation? Each of the different untranslated regions tested by reporter assays operate using different mechanisms. For example, the three *EEF1B2* transcript isoforms all share the same uORF, but alternative splicing changes the distance between the uORF and the start codon, which can influence the efficiency of downstream initiation (*Grant et al., 1994*; *Hinnebusch, 2005*). Globally, we found a large difference between the 3′ UTR lengths of poorly- versus well-translated isoforms by querying the prevalence of features likely to influence translation (*Figures 3* and *5*). There are at least three possibilities for how 3′ UTRs could influence translation. First, miRNA-mediated regulation targets the 3′ UTR (*Mayr and Bartel, 2009*; *Sandberg et al., 2008*). Second, numerous RNA binding proteins target the 3′ UTR through AU-rich elements or other specific binding sites and are known to influence translation, mRNA decay and even protein localization (*Berkovits and Mayr, 2015*; *Szostak and Gebauer, 2013*; *Zhao et al., 2014*). Lastly, it is possible that increasing the length of the 3′ UTR influences the impact of the mRNA closed loop on translation initiation or reinitiation (*Amrani et al., 2008*; *Costello et al., 2015*). Targeted work on specific 3′ UTR isoform sets could test the different possibilities for 3′ UTR-mediated translational control. We additionally demonstrated isoform-specific translational control for select 5′ leaders (*Figures 1* and *5*) as was shown genome-wide in yeast, where 5′ leaders are short enough to directly sequence (*Arribere and Gilbert, 2013*), and find that globally uORFs can both up- and down-regulate translation (*Figure 3*). However, we find a surprisingly minor global dependence of other 5′ leader features tested on translation. The mechanisms of translational control of transcript isoforms therefore follow some general trends but are likely highly idiosyncratic; these data empower investigations into the mechanisms underlying transcript-specific translational control for thousands of human genes.

We predicted the consequences of transcript isoforms observed during human embryonic development on protein production (*Figure 4*). We show that even small changes to transcripts can dramatically affect protein production (*Figure 5*), so changes at the isoform level (*Figure 4*) can considerably affect protein abundances even if the gene-level RNA expression remains similar. The application of TrIP-seq to different cell types should yield cell type-specific translation enhancer and repressor elements, contributing to the understanding of cell type-specific translational control, and augmenting our ability to precisely engineer translation in complex systems. If simpler sample

preparation is desired, it is likely sufficient to sequence the cytoplasmic and monosome fractions, as well as pooled two-four ribosome and five-eight+ ribosome fractions, as these cluster together (*Figure 1B*). Accumulation of TrIP-seq data in additional cell types may enable building a holistic model of translational control with isoform resolution in human cells.

We showed that the translation of an arbitrary gene, *Renilla* luciferase, can be controlled in a manner predicted by TrIP-seq data over two orders of magnitude (*Figure 5*) and in different cell types (*Figure 6*). These data can thus be used to select regulatory regions to control translation, without redesign of the coding sequence of the message (*Figure 7B*). It may prove superior to use 3′ UTRs to design cell type specific translation, as the repertoire of miRNAs and RNA-binding proteins that may affect translation through the 3′ UTR vary between cell types (*Figure 6B*). Indeed, in *Caenorhabditis elegans*, 3′ UTRs are sufficient to specify germline-specific expression of the attached ORF (*Merritt et al., 2008*). Even without engineering of unnatural mRNAs, it may be possible to use antisense oligonucleotides to direct splicing of poorly- or well-translated isoforms to adjust protein expression in situ (*Kole et al., 2012*), for example to downregulate the oncogenic *NAE1* gene [*Figure 5A*, *Xie et al., 2014*]), which is currently being targeted by small molecules (*Luo et al., 2012*; *Wu and Yu, 2015*). We anticipate the ability to tune the translational output of mRNA will facilitate research and therapeutic uses of designed and endogenous mRNA molecules.

## Materials and methods

### Sucrose gradient fractionation of polysome profiles

Two independently passaged, biological replicate 15 cm dishes of HEK 293T cells obtained from the University of California, Berkeley cell culture facility were grown to ~70% confluency in DMEM + 10% FBS. The cell line was authenticated by DDC Medical (Fairfield, OH) and were verified to be free of mycoplasma contamination. Cells were actively growing when harvested. The media was aspirated and replaced by PBS + 100 µg/ml cycloheximide and incubated at 37°C for 10 min. We chose cycloheximide because it induces rapid protein synthesis arrest (*Han et al., 2014*), has been successfully used in ribosome profiling of HEK 293T cells (*Sidrauski et al., 2015*), and 100 µg/ml is ~100 times higher than the concentration required to inhibit protein synthesis in reticulocytes (*Godchaux et al., 1967*). Each dish was then placed on ice, media aspirated, and replaced by ice cold PBS + 100 µg/ml cycloheximide. Cells were scraped, pelleted at $16,000\times$ g for 30 s, and re-suspended in three pellet-volumes ice cold hypotonic lysis buffer (10 mM HEPES pH 7.9, 1.5 mM $MgCl_2$, 10 mM KCl, 0.5 mM DTT, 1% Triton X-100 and 100 µg/ml cycloheximide) (*Folco et al., 2012*). After 10 min, cells were lysed on ice by ten strokes through a 26-gauge needle and nuclei were pelleted at $1,500\times$ g for 5 min. Lysate from ~15 million cells (one dish) was layered on top of triplicate 10–50% (w/v) sucrose gradients (20 mM HEPES:KOH pH 7.6, 100 mM KCl, 5 mM $MgCl_2$, 1 mM DTT and 100 µg/ml cycloheximide) made using a Biocomp Instruments (Canada) gradient master. Gradients were centrifuged for 2 hr at 36,000 RPM in a SW-41 rotor, punctured, and manually peak fractionated using real-time $A_{260}$ monitoring with a Brandel (Gaithersburg, MD) gradient fractionator and ISCO (Lincoln, NE) UA-6 detector.

### Sequencing library construction and deep sequencing

RNA was extracted from pooled technical triplicate sucrose gradient fractions by ethanol precipitation followed by acid phenol:chloroform extraction. Direct phenol:chloroform extraction was precluded by phase inversion in high sucrose fractions. RNA was then DNase treated, acid phenol:chloroform extracted, and ethanol precipitated. Cytoplasmic RNA was TRIzol extracted (Life Technologies, Grand Island, NY) and ethanol precipitated. Total RNA integrity was verified using a BioAnalyzer (Agilent, Santa Clara, CA). Ribosomal RNA was then depleted using Ribo-Zero (Illumina, San Diego, CA) and biological duplicate sequencing libraries were generated using the TruSeq RNA Sample Prep v2 kit (Illumina) without the poly-A selection steps. An equal mass (100 ng) of rRNA-depleted RNA was used as input to each individual library preparation. Libraries were verified using a BioAnalyzer and quantified using a Qubit (Life Technologies) prior to pooling for sequencing. Library insert sizes were typically ~150 ± ~25 bp. Pooled libraries were 75-bp paired-end sequenced on an Illumina HiSeq 2500 and runs of the same library in different lanes or flowcells were merged.

## Sequencing data processing and transcriptome alignment

Adapters were trimmed using Cutadapt v1.5 (*Martin, 2011*) followed by subtractive alignments against the repeatmasker (RMSK) database (retrieved from UCSC on 2/11/2015) and abundant sequences from the Illumina iGenomes project (e.g. ribosomal RNA and the mitochondrial chromosome) using Bowtie2 v2.2.4 (*Langmead and Salzberg, 2012*). Unaligned reads were then aligned to the Ensembl release 75 transcriptome using Tophat v2.0.13 with parameters "-r 5 –mate-std-dev 50 -g 100 –report-secondary-alignments" (*Trapnell et al., 2009*). Mapping percentages for each pipeline stage are presented in *Figure 1—figure supplement 3*. Transcript isoform level abundances were calculated using Cuffquant v2.2.1 (*Roberts et al., 2011*; *Trapnell et al., 2010*), normalized between samples with Cuffnorm v2.2.1, and transcripts per million (TPM) (*Wagner et al., 2012*) values were calculated according to:

$$TPM_i = RPKM_i \frac{10^6}{\sum_{g \in all\ genes} RPKM_g}$$

Biological duplicate datasets were processed independently. Quantified abundances are in *Figure 1—source data 1* and *Figure 1—source data 2*. TrIP-seq plots are available for all Ensembl GRCh37 isoforms at http://meru.qb3.berkeley.edu/tripseq.

## Hierarchical clustering of isoform distributions across the polysome profile

Cuffnorm counts for each replicate were subjected to a variance stabilizing transformation (VST) using the DESeq2 R package to correct for heteroscedasticity (*Love et al., 2014*); the VST approaches $\log_2$ for large counts but compresses low counts to suppress Poisson noise. Variance stabilized counts were averaged between replicates, filtered such that the mean across all nine samples was greater than one (roughly translating to 100 reads), and then mean subtracted to generate relative expression values. Inter-row distance was computed using Spearman's rank correlation, and hierarchical clustering was performed using the fastcluster R package (*Müllner, 2013*) with Ward's agglomeration method. The resulting dendrogram was split at increasing heights into subtrees until clusters contained similar overall trends, generating the large clusters presented in *Figure 2*. The samples were clustered similarly except with Euclidean distance and complete agglomeration. The R packages magrittr, dendextend *Galili, 2015*, and ggplot2 *Wickham, 2009* were used to generate figures shown and are available through CRAN. Transcript isoforms in each cluster are in *Figure 2—source data 1*.

## Clustering of TrIP-seq samples

The intersample clustering in *Figure 1B* was performed by hierarchically clustering the VST-transformed isoform-level counts as above between samples. The statistical significance of the resulting clustering was analyzed by measuring the Jaccard distance between clusterings of these data after subjecting the data to subsampling by bootstrap, jittering (adding random noise to each point), or replacing random points by noise using the R package fpc. The mean Jaccard distances of 100 such subsamplings as well as the average are presented in Figure *Figure 1—figure supplement 1G* S1G; a Jaccard distance $\geq$.75 is considered a stable cluster by the R package fpc.

## Cloning of untranslated regions and in vitro transcription

Isoform-specific 5′ and 3′ untranslated regions were amplified from anchored oligo-dT primed cDNA libraries from HEK 293T cells and Gibson cloned (*Gibson et al., 2009*) into a vector based on pUC57 containing *Renilla* luciferase and a synthetic polyA$_{60}$ tail (pA60; *Fukaya and Tomari, 2011*). Gibson cloning was performed such that untranslated regions were precisely cloned next to the ATG but contained two guanosines as the 5′-most nucleotides for T7 transcription for 5′ leaders, or the stop codon used by the isoform, and the six nucleotides CTGCAG at the 3′ end of the 3′ UTR immediately preceding the polyA$_{60}$ tail for 3′ UTRs. The differences between cloned isoforms are shown in *Figure 5—figure supplement 1*, which was generated using IGV (*Thorvaldsdottir et al., 2013*). The entirety of all untranslated regions was verified using dideoxy sequencing. Transcription templates were generated by PCR using Phusion polymerase (NEB, Ipswich, MA), size verified by gel electrophoresis, and gel purified. Transcription was performed using T7 polymerase with 1 μg template in a

buffer containing 7.5 mM each NTP, 1 µg pyrophosphatase (Roche, Pleasanton, CA), 30 mM DTT, 35 mM MgCl$_2$, 2 mM spermidine, 0.01% Triton X-100 and 30 mM Tris pH 8.1 for four hours at 37°C followed by DNase treatment with RQ1 RNase-free DNase (Promega, Madison, WI) for 30 min. Transcription products were purified by ethanol precipitation and a Zymo (Irvine, CA) Clean & Concentrator column, followed by simultaneous capping using Vaccinia capping enzyme (NEB) and 2′-O-methylation (NEB). Capped products were purified using a Zymo Clean & Concentrator column, followed by size verification on glyoxylated samples (Ambion, Foster City, CA) using an agarose gel, and full-length 3′ UTR-containing RNAs were gel purified from an agarose gel (Zymo).

## Transfections and luciferase assays

The concentration of RNAs containing 5′ untranslated regions was determined using A$_{260}$, which was normalized by the intensity of the full-length product on an agarose gel, and then by the molar ratio of the construct to empty pA60. Molar-adjusted amounts of each RNA relative to 100–200 ng of pA60 were transfected into three technical triplicate wells of ~50% confluent cells in a 96-well plate using the TransIT-mRNA reagent (Mirus, Madison, WI). The concentration of RNAs containing 3′ UTRs was determined using a Qubit RNA HS assay (Life Technologies), normalized for the molar ratio of the construct to empty pA60, and molar-adjusted amounts of RNA relative to 7 ng of pA60 were transfected using TransIT-mRNA. A reduced amount of the 3′ UTR RNAs was used due to decreased yield of the longest 3′ UTRs. HEK 293T cells were grown in DMEM + 10% FBS, Hep G2 cells were grown in EMEM + 10% FBS, MCF7 cells were grown in DMEM:F12 + 10% FBS, A549 cells were grown in F12-K media + 10% FBS, and K-562 cells were grown in RPMI media + 10% FBS. Cells were harvested after ~18 hr (*Figure 5*) or 2 hr (*Figure 6*) and *Renilla* luminescence was measured (Promega).

## Tabulation of isoform features

The Ensembl release 75 annotation set from the Illumina iGenomes project was first decomposed into 5′ leader, start codon, CDS, 3′ UTR, and whole-transcript regions. Length, GC-content, and number of exons were computed directly. Cytoplasmic expression and the median expression from the 80S through the eight+ ribosome fraction were extracted from TrIP-seq data. Transcript halflife data were derived from HeLa cell measurements (*Tani et al., 2012*). The structure was computed using RNALfold from the ViennaRNA package (*Lorenz et al., 2011*) in a 75-nt window. Codon usage statistics were downloaded from http://www.kazusa.or.jp/codon/ on 11/20/2014 and the minimum codon frequency is the average of codon usage across a five-codon window. The fraction of AU-elements is calculated as the percentage of the 3′ UTR that is of repeating A or U nucleotides for more than 5nt in a row. TargetScan 6.2 scores (*Garcia et al., 2011*) were downloaded from http://targetscan.org and parsed for the properties indicated, and an identical comparison was performed after filtering the miRNA list by those expressed in HEK 293T cells (*Ender et al., 2008*). All feature tabulation was performed using custom Python programs, which are available through GitHub at https://github.com/stephenfloor/tripseq-analysis.

## Effect size measurement between isoform feature distributions

Isoforms belonging to the same gene present in different clusters were compiled, yielding gene-linked isoforms. For example, if the gene A has isoform 001 in cluster one and isoforms 002 and 003 in cluster two, features of isoform 001 are added to the cluster one set and features of 002 and 003 are added to the cluster two set and 001–002 as well as 001–003 would be gene-linked isoforms. Features in each set were then compared for statistical significance using the Mann-Whitney U test and visualized as empirical cumulative distribution functions (*Figure 3—figure supplement 1A*). The effect size was then computed between distributions using Cliff's *d*, which is a nonparametric, dimensionless measure of the distance between distributions (*Cliff, 1993*). Cliff's *d* is a measure of the number of times that a point $x_i$ in one distribution is greater than all points $x_j$ in the second distribution, or

$$d = \frac{\#(x_i > x_j) - \#(x_i < x_j)}{mn}$$

where # denotes the number of times, the two distributions are of sizes $n$ and $m$, and $x_i$ and $x_j$ are items of the two sets. Cliff's *d* is also related to the Mann-Whitney U statistic, by

$$d = \frac{2U}{mn} - 1$$

Cliff's $d$ and the boostrap confidence intervals shown in *Figure 3* were computed using the R package orddom, which is available through CRAN.

## Ribosome profiling comparison and TrIP-seq polysome count calculation

Ribosome profiling was performed in HEK 293T cells (*Sidrauski et al., 2015*). These data were downloaded from the NCBI (GEO: GSE65778) and reprocessed by subtractive Bowtie 2 alignment to rRNA and RMSK sequences, and then mapped onto the human transcriptome using Tophat, as for the TrIP-seq data. Aligned reads were quantified at the gene and isoform level using Cuffquant onto the Ensembl release 75 transcriptome and normalized with Cuffnorm as above, to facilitate direct comparison to the TrIP-seq data. The number of reads derived from polysomes for TrIP-seq data was computed as the sum of read counts for fractions containing two to eight+ ribosomes multiplied by the number of ribosomes in each fraction:

$$\text{TrIP} - \text{seq polysome counts} = \sum_{i=1}^{7} i\,n_i + 24n_8$$

where $n_i$ is the number of reads in the $i$th polysome fraction averaged for both biological replicates. The factor of 24 is applied to the last fraction based on a ceiling of ~40 ribosomes per transcript which then leads to an average of 24 ribosomes in the eighth peak (40 + 8 / 2 = 24). The number of ribosomes in the final peak is not directly measured by the sucrose gradients used *Figure 1—figure supplement 1A*).

## Reprocessing of embryo sequencing data

RNA sequencing datasets of human preimplantation embryos (*Yan et al., 2013*) were downloaded from the NCBI (GEO: GSE36552). Reads were converted to FASTQ using fastq-dump (NCBI) and then processed as for TrIP-seq data by adapter trimming with Cutadapt, subtractive alignment to the RMSK and abundant sequences, and transcriptome alignment with Tophat to Ensembl release 75. Transcript abundances were calculated using Cuffquant and normalized with Cuffnorm. Data from individual cells were processed independently and averaged for each stage to generate the data shown in *Figure 4* and *Figure 4—figure supplement 1*. The data were reprocessed for consistency to facilitate direct abundance comparisons.

## RT-PCR

Cytoplasmic lysate was fractionated using a sucrose gradient as for the RNA-seq libraries and RNA was extracted from each fraction. Libraries of cDNA were then synthesized from these fractions using random primers (Applied Biosystems) from an equal amount of RNA as measured using a Qubit (Life Technologies; RNA HS assay). Template RNA was removed using RNase H treatment (NEB) and cDNAs were purified over an oligo cleanup column (Zymo). PCR was then performed for 25 (*ACTB*) or 30 (*EEF1B2, ATF4*) cycles with gene-specific primers (below) using Taq Titanium (Clontech) using an equal amount of cDNA input into each PCR, as measured using a Qubit (ssDNA assay). Reactions were then run on a 1% agarose gel and stained with SYBR-Gold (Life Technologies). Gels are representative of three biological replicates.

Primers used for RT-PCR are:
ACTB-forward AGAGCTACGAGCTGCCTGAC
ACTB-reverse AGCACTGTGTTGGCGTACAG
EEF1B2-forward TTCCCGTCATCTTCGGGAGCCGT
EEF1B2-reverse CTTTTCAGGTCTCCGAAACCCATGG
ATF4-forward GGCTCTGCAGCGGCAACCCC
ATF4-reverse CGACTGGTCGAAGGGGGACA

## Quantitative RT-PCR

RNA was extracted from polysome fractions as for TrIP-seq. qRT-PCR was performed using the SuperScript III Platinum SYBR Green One-Step kit (Life Technologies) with 1ng input RNA in a 20 ul

reaction volume. $C_T$ values were converted to fold changes over the cytoplasmic abundance and plotted.

Primers used for qRT-PCR are:
EEF1B2-001-fwd AGCCGTGGAGCTCTCGGATA
EEF1B2-001-rev AGCTCTTGTCCGCCAGGTAA
EEF1B2-003-fwd TCCAAACGCAACGAAAGGTCC
EEF1B2-003-rev CGCCGGACCGAAGGTTAAAG
EEF1B2-201-fwd AGCCGTGGAGCGTGGG
EEF1B2-201-rev TCGGCTGTATCCGAGAGCTG
SRSF5-002-fwd GACCCCGTCCGGTAGGAAGTACTAGCC
SRSF5-002-rev CAATATCTCTTATCCGTCCATATCCC
SRSF5-005-fwd GGTGAGTGGCTCACTTTGAGGGCAAG
SRSF5-005-rev CGAATCAACTGCGCTCATTAGACGC

## Gene Ontology analysis

The DAVID server was used to calculate gene ontology (GO) terms for biological processes (BP) associated with the individual clusters (*Jiao et al., 2012*). Transcripts associated with each cluster were input into DAVID and GO BP terms with a Benjamini-corrected p-value of <0.05 were tabulated.

## Accession numbers

Raw sequencing reads for all samples are available through the NCBI via the GEO Accession ID GSE69352.

## Acknowledgements

We thank Y Bai, P Kranzusch, A S Y Lee, E. Montabana, M O'Connell, D Rio and A Tambe for critical reading of the manuscript, N Ingolia, S Iwasaki, L Lareau and members of the Doudna lab for helpful advice, and A Fischer, K Condon and M Chung for technical assistance. We thank S Oh and Yoon-Jae Cho for sharing data prior to publication. This work used the Vincent J. Coates Genomics Sequencing Laboratory at UC Berkeley, supported by NIH S10 Instrumentation Grants S10RR029668 and S10RR027303.

## Additional information

### Funding

| Funder | Author |
|---|---|
| Howard Hughes Medical Institute | Stephen N Floor<br>Jennifer A Doudna |
| Helen Hay Whitney Foundation | Stephen N Floor |

The funders had no role in study design, data collection and interpretation, or the decision to submit the work for publication.

### Author contributions

SNF, Conceptualization, Methodology, Investigation, Software, Writing - Original Draft, Funding Acquisition; Writing - Review & Editing; Supervision, Conception and design, Acquisition of data, Analysis and interpretation of data, Drafting or revising the article; JAD, Writing - Original Draft, Funding Acquisition, Supervision; Writing - Review & Editing, Drafting or revising the article

## Additional files

### Supplementary files

• Supplementary file 1. RNA sequences for untranslated regions fused to luciferase (*Figure 5*).

## Major datasets

The following datasets were generated:

| Author(s) | Year | Dataset title | Dataset URL | Database, license, and accessibility information |
|---|---|---|---|---|
| Stephen Floor, | 2015 | Human isoform-specific translational control uncovered by Transcript Isoforms in Polysomes sequencing (TrIP-seq) | https://www.ncbi.nlm.nih.gov/geo/query/acc.cgi?acc=GSE69352 | Publicly available at the NCBI Gene Expression Omnibus (Accession no: GSE69352). |

The following previously published datasets were used:

| Author(s) | Year | Dataset title | Dataset URL | Database, license, and accessibility information |
|---|---|---|---|---|
| Sidrauski C, McGeachy A, Ingolia N, Walter P, | 2015 | The Small Molecule ISRIB Reverses the Effects of eIF2$\alpha$ Phosphorylation on Translation and Stress Granule Assembly | http://www.ncbi.nlm.nih.gov/geo/query/acc.cgi?acc=GSE65778 | Publicly available at the NCBI Gene Expression Omnibus (Accession no: GSE65778). |

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
