## [Decision Letter]

[Editors’ note: this article was originally rejected after discussions between the reviewers, but the authors were invited to resubmit after an appeal against the decision.]

Thank you for submitting your work entitled "Tunable protein synthesis by transcript isoforms in human cells" for consideration by *eLife*. Your article has been reviewed by two experts in the field, and the evaluation has been overseen by a Reviewing Editor and James Manley as the Senior Editor. Our decision has been reached after extensive consultation between the Reviewing Editor and the two reviewers. Based on these discussions and the individual reviews below, we regret to inform you that your work cannot be considered further for publication in *eLife*.

Both reviewers felt that the data were overall of very high quality and the general approach an interesting (albeit not novel) one to get at isoform specific information on translational efficiency. In this regard, the TrIP-Seq approach as presented is basically identical to previous experimental designs (Sterne-Weiler et al. 2013; Maslon et al. 2014; Jingyi Hou et al. 2015). Nevertheless, the degree of novelty of the approach was not the key factor in the decision. Further, the reviewers were impressed with the reporter data (in the original cell type HEK293) that provided strong evidence for differential translation of isoforms, as suggested and then followed up from the initial observations from the global approach. The reporter data led to an interesting model where 5' UTR leader sequences contribute generally to overall translational efficiency. The authors further proposed a model wherein 3' UTRs would be generally more important in specifying cell-specific regulation, based on reporter analysis in several different cell types. However, the reviewers felt that these data did not provide sufficient breadth to justify the broad conclusions that were made and felt that further global analysis in different cell types would have provided this support. Overall, the reviewers concluded that this high quality analysis failed to sufficiently support the broad claims made concerning isoform-specific translation patterns but rather represented the beginnings of such an analysis. In the absence of broader exploration of this topic in multiple cell types (and a comparative analysis), the study falls short of its claims and thus is not appropriate for publication in *eLife*.

Reviewer #1:

In this paper Floor and Doudna apply RNA-Seq to identify the presence of previously annotated isoforms in polyribosome fractions isolated from HEK 293T cells. Then the authors classify the identified isoforms according to their distribution along the sucrose gradient and select 7 genes for confirmation of their results. The work is well done and the experimental confirmation goes beyond previous works, however the "TrIP-Seq" approach is basically identical to previous experimental designs (Sterne-Weiler et al., 2013; Maslon et al., 2014; Jingyi Hou et al., 2015). Taking that in consideration I think that the novelty of the manuscript it is not sufficient to merit its publication in *eLife*, but rather in a more specialized journal.

Major comments and concerns:

1) In general the claims of the manuscript seem to go beyond what it is actually demonstrated. All the presented data is based in an experiment performed in HEK 293T cells. And the confirmation in other cancer cell lines is restricted to a handful of UTRs. To produce "an atlas of isoform-specific translation patterns" as the authors claim it would be necessary to apply their method in different cell lines and conditions where translation regulation is expected. However that is clearly out of the scope of this first manuscript. To avoid misleading the readers, I would advise the authors to town down their claims and Discussion.

2) In paragraph three, subheading “TrIP-seq measures transcript isoform-specific translation in human cells”, the authors compare their approach with other measures of polyribosome association and translation. It is clear that their method measures polyribosome abundance, however the presented comparisons do not demonstrate that they measure translation. For example in Figure 1, neither TrIP-Seq nor ribosome profiling counts are corrected by mRNA abundance. Thus the observed correlation could be simple due to mRNA abundance (and not the translation efficiency). I would not expect their method to obtain a perfect correlation with ribosome protection or protein abundance, however it could be expected to be significantly greater than the one measured by mRNA abundance alone.

3) The authors make special emphasis in the distinction between high and lowly- ribosome associated transcripts (e.g. paragraph one, subheading “TrIP-seq measures transcript isoform-specific translation in human cells” and paragraph two subheading “3′ UTRs and introns drive isoform-specific polysome association”). The authors claim that isoforms "with more exons tend to be better translated" and that that is not caused by transcript length. However in Figure 3 and supplement, it can be observed that the length of the CDS could explain part of those differences. If that were the case, it would lead to the trivial observation that genes encoding longer proteins can potentially accommodate more ribosomes in their coding regions. That will cause those isoforms to be present in higher polyribosome fractions (independent of their translational potential). This effect would also potentially affect the clustering shown on Figure 2, which would be affected both by the CDS length and the translation efficiency (polyribosome association).

4) It is not clearly defined how the authors define biological replicates. If the authors refer to different batches of cells grown independently and subjected to different sucrose fractionationations they should specify that in the Methods section.

*Reviewer #2:* There is a very interesting finding that seems very likely correct. However, the manuscript as presented has many issues that should be rigorously addressed, as noted below:

1) Abstract: There is no value in stating that the relationship of RNA and protein levels is 'controversial'; that's not really scientific. Indeed, this is the case either or both because there is too little data, data are wrong, or the right type of data have not been collected. I think this manuscript makes a great case for the last of these. I.e., prior work (arguably) has been too crude to ferret out or isolate different factors responsible. Here, having the gene the same but changing the 5' or 3' UTR is a much better controlled experiment to sort out these effects. I know that wasn't the authors' intention – they wanted to discover these effects – but serendipity has given them a great controlled experiment as well. My guess is that this is why they're able to get so much more information.

2) Abstract: A suggestion (strongly) not to use the term rheostat. Analogies can be helpful, distracting, or misleading. Here at a minimum the rheostat is not a physical analogy. Presumably the effects arise from molecular interactions – and the results lead to the neat model that 3'UTRs are cell type specific because different cells have different proteins expressed (providing additional evidence in support of the 3' UTR as hosting a complex panoply of proteins; the constancy of 5'UTR effects suggests that these arise from something common to all cells –perhaps the ribosome and translation initiation machinery.

3) Introduction: The first paragraph is considerably under-documented. How well established are 'diverse translational control elements'? There are certainly a number of specific cases, but is their number and breadth clear? And the three examples given are maybe not the most clear or general. While one can't have a review here, an opening perspective that is both broader and contains important details would provide the scholarly context for this work.

4) Introduction: The authors point out that there is not a technique to look at isoform specific translation. They then take nearly a page to note how 'ribo-seq' cannot do this. They then reveal that sucrose (polysome) gradients, followed by deep sequencing, can do this. But of course polysome fractionation is a very old technique. And it is not until the Results (paragraph one, subheading “TrIP-seq measures transcript isoform-specific translation in human cells”) that we learn that Arava et al. did polysome fractions for all yeast genes – at that time using microarrays. And microarrays could be designed to look at isoforms, or looked at another way, the technical extension of this work from the existing technologies is a modest step, but not presented in the appropriate historical context. While this is common, it is not needed. The biology and importance of the question stands for itself. So the authors’ main advance is not a technique – but rather the identification of a (very) important question and unmet need, and the expert execution of addressing this question. (As a related aside, if one goes back to the original Ignolia et al. Science paper introducing ribo-seq, there the method has held – and been improved – and has been very important, but each of the scientific conclusions drawn was either made previously (by Arava et al.) or later shown to be incorrect. That paper still deserved the wide audience it got, as does this one, but let's work hard to get the intellectual and scholarly history right, whether or not it involves prior work that I was associated with.)

5) Subsection “TrIP-seq measures transcript isoform-specific translation in human cells” and Figure 1: It is not clear that had Figure 1 not recapitulated the gradient order that it would have meant that there is something amiss in the experiment.

While the clustering 'broke' between poly 4 and 5, this is not a statistically tested or rigorous result, and there are many possible origins. So the conclusion (from this observation at least) that the data can be considered in terms of low and high polysomes is not supported. One variable to consider is that short genes (which are highly abundant) will by necessity cluster in the low region. Thus, that statement “This suggests that in cells, transcript isoforms are predominantly poorly- or highly-ribosome associated, causing low polysomes to cluster away from high polysomes” is an over-interpretation of the data that are presented.

Technical: Was the clustering done on all reads or on a per-gene basis?

Technical: How is TPM (transcripts per million) determined? What controls for the validity of this measure are there?

Technical: What controls for rapid freezing of translation upon cycloheximide addition were performed? The extent of reads near the 5' end of the ORF should be a measure of this and should be noted and documented.

Figure 1: While it is true that noncoding are preferentially in low polysome fractions, they ARE in those fractions, and the difference between low and high is only 2x. Is there reason to be concerned by this? (The statement “Protein coding genes and isoforms are found across the polysome, but noncoding and lncRNA genes are predominantly in the low polysome fractions indicating that they are generally weakly ribosome associated or in other large macromolecular complexes (Figure 1 and Figure 1—figure supplement 1)” seems overstated given the 2x differential.)

Figure 1: The description is inadequate – what are the points?

6) “Agreement between the isoform-specific polysome profiles and RT-PCR conducted from polysome fractions verified the accuracy of the isoform-level quantifications (Figure 1)”. Figure 1 does not show even qualitative agreement between RT-PCR and TrIP-seq results. This gel and the one in the supplement should be quantitatively compared to the trip-seq data – quantitative PCR is probably needed here.

7) “Weaker amplification of the longer isoform (EEF1B2-003) in high polysomes may be due to PCR bias towards smaller amplicons (Walsh et al., 1992), and replicate experiments show EEF1B2-003 present in high polysomes (Figure 1—figure supplement 1)”. The comment indicates less than robust data agreement, and a potential reason for this. That suggests that additional experiments (like shorter PCR reads) should be done to determine if the expected agreement is indeed observed.

8) Subheading “TrIP-seq measures transcript isoform-specific translation in human cells”. Where the trip-seq replicates completely independent biological replicates? If not, this conclusion cannot be drawn.

9) “Therefore, TrIP-seq and ribosome profiling agree at the gene level but diverge at the isoform-level, likely due to difficulty in mapping the very short profiling reads at the isoform level, or the influence of untranslated regions, which are invisible to profiling”. As I understand it there are two models for the differences between full ribosome profiles and that for the isoforms – noise from fewer reads or 'real' effects. This isn't so clear from the wording here, and seems like it needs to be addressed here – since one is an artifact and the other is biology.

10) “We then compared TrIP-seq data to copy number estimates from HeLa cells, and find that the two datasets correlate well (RS = 0.55, Figure 1—figure supplement 1, (Kulak et al., 2014)”. Rs – 0.55 is viewed as good, but similar correlation coefficients are viewed as bad in other contexts. Maybe instead of saying correlate 'well' we need to better understand the factors that affect this correlation. Figure 1—figure supplement 1 shows a spread in Hela copy number (protein #) that looks much broader than the spread in trip-seq polysome counts. Visually I don't think a neutral reader would say that trip-seq data account well for the protein levels.

Importantly, the rest of this paragraph is logically unclear – statements are made but the underlying evidence for those statements seems an opinion rather than a reasoned series of considerations. The final statement is strong: 'we conclude…' – but the basis for that conclusion does not appear strong or minimally is not clearly laid out. And these are the crucial data for that paper.

11) Subheading “An atlas of human isoform-specific translation patterns”, Figure 2; clustering. Clustering is statistically ambiguous, ultimately, and these data are pretty 'flat' – i.e., there aren't many very strong features to cluster by. But maybe more centrally since clustering is a somewhat arbitrary (though often useful) transformation – e.g., there is no natural or set number of clusters for a dataset- it is not clear that one what's to be making conclusions – rather than just organizing data – from clusters. The authors conclude that it's 'crucial to fractionate the polysome profile.' I think that's strongly expected a priori, but I don't see how that was strongly tested or established from the cluster analysis – other than to note that there are some differences in profiles.

12) Paragraph two, subheading “An atlas of human isoform-specific translation patterns”. The conclusions in the paragraph seem too strong from the data. The authors above said that they don't know why ncRNAs are in the polysomes – which is reasonable. But here they make conclusions about decay and pioneering rounds – these arguments are likely not germane and do not seem to be supported by the data or logic but rather seem to be 'fits' to existing ideas and models.

13) “Surprisingly, the poorly translated cluster 7, which has ~9,000 individual transcripts…”. It is not clear to me that seeing ~10K sequences with introns means that they are being translated. That's one model. The other, stated earlier in the manuscript, is that they are in other large complexes.

14) “In sum, clustering of isoform abundance distributions across polysomes immediately provides insight into the diverse patterns of ribosome association by individual isoforms and transcript types in cells, and reveals intron-rich clusters of transcripts associated with polysomes that escape nuclear detention”. For similar reasons, I don't see how this conclusion is reached, relative to other possibilities.

15) Subheading “3′ UTRs and introns drive isoform-specific polysome association”. Here the authors are using the clusters to explore and identify possible trends. This seems like a powerful exploratory use of the clusters.

16) “We find that longer coding sequences and highly abundant versions of gene-linked isoforms are biased towards high polysomes (Figure 3 and Figure 3—figure supplement 1)”. This can be because short genes can't hold as many ribosomes – or they reach the end sooner and thus fall off sooner.

17) Figure 3, subheading “3′ UTRs and introns drive isoform-specific polysome association”. A number of interesting points are raised and discussed. The bottom line at the end of this section seems vaguer than one would like – e.g., what is really established vs what might be and what might have other explanations. More basically, is the best way to analyze these data from pre-clustered groups or, since the top and bottom fractions are compared, to use a numerical ratio of those – so that one analyzes a continuous rather than discretized variable (as this distorts error).

18) Subheading “Predicting translation changes during preimplantation human development”. What does 'reprocessed' mean? It's not clear from what's stated what this dataset is? Presumably it is not polysome data – so presumably it's seeing whether gene isoforms change in abundance (lit data) that have isoform specific differences (this data).

19) Paragraph two, subheading “Predicting translation changes during preimplantation human development”. Seems like general conclusion about use of trip-seq from one example – and even that example doesn't seem so striking, and given that there are 1000's of data, the chance of a small number having seemingly statistically significant effects seems high. This might be considered as some form of multiple hypothesis testing.

20) It seems like the best independent test of the data and models is predicting new effects, and that's what this section does. The key statement seems to be “Not only do these data provide strong evidence that the output of heterologous reporters can be predictably tuned by untranslated regions, it additionally validates that the polysome profiles observed are connected to translational output”, stating that the results have predictive value. But the comparison is complex – i.e., showing polysome profiles. There are standard ways to convert polysome occupancy into protein production rate. While imperfect, that's the basis for earlier stuff in this and other work. So what seems clearest and least biased would be to make quantitative predictions for several – preferably 10-20, variants, and compare the output to those quantititave prior predictions. Figure 6 (below) suggests that this should be doable for 5' leaders but not 3' UTRs.

21) Discusssion. Where does the value of two orders of magnitude come from?

While no large translational regulation was found in prior work with 3000 highly expressed genes, are there not examples of translational regulation of genes in humans? Or in related species with conserved genes?

Paragraph two, Discussion. As noted above, the statement that trip-seq provides a 'faithful measure' may not be well supported by the analyses presented.

It seems that by far the strongest support for the manuscripts conclusions is the 'engineering', though that can be improved as noted above.

22) An interesting test would be to demonstrate/test isoform-specific effects of a miRNA.

---

## [Author Response]

[Editors’ note: the author responses to the first round of peer review follow.]

Thank you for reviewing our paper on transcript isoform specific translation in human cells. We were pleased that the reviewers thought that the work is "well done", presents a "very interesting finding" and that our paper deserves as wide of an audience as the pioneering work of Ingolia, 2009 (Science). The reviewers were also "impressed with the reporter data", where we show that isoform-specific sequences derived from TrIP-seq data can be used to control translation of unrelated genes over 100-fold.

We believe that the work conducted in HEK 293T cells represents an advance in our understanding of how the diverse human transcriptome relates to protein translation in cells that warrants a broad audience. No prior study has observed the extent of transcript isoform specific translational control uncovered by our work in any system. We are grateful for the reviewers’ comments, which have helped us to clarify the significance of the work.

New additional data or analyses to include (one experiment and eight analyses):

a) Author response Figure 1: polysome association of noncoding transcripts and associated ribosome profiling protected fragments;

b) Author response Figure 2: retained intron transcripts found in polysomes harbor ribosome profiling protected fragments, suggesting they are not simply associated with other large macromolecular complexes;

c) Correlation between translation efficiency as measured by TrIP-seq and ribosome profiling (regarding Figure 1);

d) Quantitative RT-PCR across polysome profiles to compare to the TrIP-seq data (regarding Figure 1);

e) Bootstrap analysis of inter-fraction clustering (regarding Figure 1);

f) A correlation between TrIP-seq data and HEK 293T proteomics (regarding Figure 1);

g) An analysis of the coding potential of retained intron sequences (regarding Figure 2);

h) Ribosome density analyses to normalize for CDS-length dependent effects in analyzing transcript isoform specific features (regarding Figure 3);

i) Quantitative comparisons between the predicted protein output and luciferase measured (regarding Figure 5).

Reviewer #1:

*In this paper Floor and Doudna apply RNA-Seq to identify the presence of previously annotated isoforms in polyribosome fractions isolated from HEK 293T cells. Then the authors classify the identified isoforms according to their distribution along the sucrose gradient and select 7 genes for confirmation of their results. The work is well done and the experimental confirmation goes beyond previous works, however the "TrIP-Seq" approach is basically identical to previous experimental designs (Sterne-Weiler et al., 2013; Maslon et al., 2014; Jingyi Hou et al., 2015). Taking that in consideration I think that the novelty of the manuscript it is not sufficient to merit its publication in eLife, but rather in a more specialized journal.*

Polysome profiles have indeed been used previously to assess gene expression levels. However, no prior study has observed the extent of transcript isoform specific translational control uncovered by our work in any system. Prior work has focused on gene-level analyses (Maslon, Hou), isolated splicing events (Sterne-Weiler), or transcript termini (Arribere, Spies), without reconstructing transcript isoforms and without analyzing effects within the polysome profile itself. Our work clearly shows that transcript isoforms of the same gene can be associated with different numbers of ribosomes (Figure 2), which is driven by diverse sequence features of isoforms (Figure 3), and that isoform-specific sequences that alter ribosome association by TrIP-seq can be used to control translation of orthogonal genes in a manner predicted by the sequencing data with a dynamic range of at least 100-fold (Figure 5). Therefore, the data from HEK 293T cells present fundamental insights into transcript isoform specific translation in humans that remain unappreciated. These data and conclusions are new relative to previous work and are of interest to researchers in many fields including translation, splicing, transcription, and cancer biology, where the transcriptome is frequently altered. Furthermore, since we show that isoform-specific sequences can control translation of orthogonal genes in a predictable manner, researchers in mRNA therapeutics or synthetic biology can mine these data for sequences that confer graded translation responses in cells.

*Major comments and concerns:*

*1) In general the claims of the manuscript seem to go beyond what it is actually demonstrated. All the presented data is based in an experiment performed in HEK 293T cells. And the confirmation in other cancer cell lines is restricted to a handful of UTRs. To produce "an atlas of isoform-specific translation patterns" as the authors claim it would be necessary to apply their method in different cell lines and conditions where translation regulation is expected. However that is clearly out of the scope of this first manuscript. To avoid misleading the readers, I would advise the authors to town down their claims and Discussion.*

We have revised the manuscript to better match our claims and conclusions with the data presented. Although it would be fascinating to apply TrIP-seq to other systems where differential translational control is expected, we agree with the reviewer that this is beyond the scope of this paper. We will instead highlight the rich set of isoform-specific translational control features observed in 293T cells, and suggest the potential for some of these features to exist in other cell types, rather than concluding that there are broad similarities between cell types. We note that while this manuscript does not describe translation in different biological conditions, it still presents a rich resource of isoform-specific translation patterns in human cells.

*2) In paragraph three, subheading “TrIP-seq measures transcript isoform-specific translation in human cells”, the authors compare their approach with other measures of polyribosome association and translation. It is clear that their method measures polyribosome abundance, however the presented comparisons do not demonstrate that they measure translation. For example in Figure 1, neither TrIP-Seq nor ribosome profiling counts are corrected by mRNA abundance. Thus the observed correlation could be simple due to mRNA abundance (and not the translation efficiency). I would not expect their method to obtain a perfect correlation with ribosome protection or protein abundance, however it could be expected to be significantly greater than the one measured by mRNA abundance alone.*

We appreciate that mRNA abundance could influence this correlation. However, the abundance of mRNA within a polysome or protected ribosome fragments is generally a measure of translation, and is what both TrIP-seq and ribosome profiling seek to measure. Thus, the most direct method to test for agreement between the two methods is through comparing polysome/protected fragment abundances directly without additional confounding variability by dividing by RNA-seq measurements.

To measure the correlation between TrIP-seq and ribosome profiling independent of mRNA abundance, we computed the ribosome profiling translation efficiency (TE) using standard methods. We then divided the weighted sum of polysome abundance of the TrIP-seq data as described in the manuscript by the cytoplasmic abundance, yielding a TrIP-seq TE. The correlation between the ribosome profiling and TrIP-seq TE measurements is RS = 0.24, which we have added to the manuscript in paragraph three, subheading “TrIP-seq measures transcript isoform-specific translation in human cells”. Possible sources of this weak correlation include different biological conditions (e.g. cell confluence) and sequencing formats, especially since Sidrauski et al used a linker-ligation based prep while this work uses random primed reverse transcription. We compared the data of Sidrauski et al. to an unpublished ribosome profiling dataset in 293T cells from the Cho lab and find that ribosome protected fragments correlate with RS = 0.78, while TE has a substantially lower correlation of RS = 0.41, suggesting biological or technical variability disproportionately affects TE. Therefore, TrIP-seq polysome counts correlate with ribosome profiling protected fragments similarly to other profiling experiments, while TE appears to be exquisitely sensitive to inter-experiment variability.

To address the relationship between TrIP-seq versus RNA-seq and protein abundance, we compared the weighted TrIP-seq polysome counts to 293T protein abundances, obtaining a correlation of 0.57 (Figure 1—figure supplement 1), while RNA-levels alone exhibit a weaker correlation of 0.45 (not shown), indicating TrIP-seq is superior to RNA-seq when predicting protein abundance. We thank the reviewer for motivating these analyses.

*3) The authors make special emphasis in the distinction between high and lowly- ribosome associated transcripts (e.g. paragraph one, subheading “TrIP-seq measures transcript isoform-specific translation in human cells” and paragraph two subheading “3’ UTRs and introns drive isoform-specific polysome association”). The authors claim that isoforms "with more exons tend to be better translated" and that that is not caused by transcript length. However in Figure 3 and supplement, it can be observed that the length of the CDS could explain part of those differences. If that were the case, it would lead to the trivial observation that genes encoding longer proteins can potentially accommodate more ribosomes in their coding regions. That will cause those isoforms to be present in higher polyribosome fractions (independent of their translational potential). This effect would also potentially affect the clustering shown on Figure 2, which would be affected both by the CDS length and the translation efficiency (polyribosome association).*

We thank the reviewer for noting that the CDS length strongly drives polyribosome association, and now explicitly discuss this in the text. We have analyzed the ribosome density across transcripts to explore ribosome association of transcripts independent of the coding sequence length. We find that gene-linked isoforms in low polysome clusters generally have lower ribosome density than those in high polysome clusters (Figure 3). Therefore, the clustering in Figure 2 is not solely due to CDS length. Furthermore, the correlation between CDS length and 3’ UTR length is positive 0.3 (not shown), suggesting that shorter CDSs on average have shorter 3’ UTRs, while Figure 3 shows short CDSs are associated with few polysomes but short 3’ UTRs are associated with many polysomes. We also note that all the reporters tested in Figure 5 and Figure 6 were selected from gene-linked isoforms bearing nearly identical CDS lengths to mitigate this effect (Figure 5—figure supplement 1). As these reporters show considerable isoform-specific translation, there is clearly signal in the data beyond CDS-length effects.

*4) It is not clearly defined how the authors define biological replicates. If the authors refer to different batches of cells grown independently and subjected to different sucrose fractionationations they should specify that in the Methods section.*

We regret that this was not clear and have updated the Methods section. Biological replicates refer to separate dishes of cells that were grown, lysed, fractionated and used to prepare sequencing libraries in completely independent experiments.

Reviewer #2:

There is a very interesting finding that seems very likely correct. However, the manuscript as presented has many issues that should be rigorously addressed, as noted below:

*1) Abstract: There is no value in stating that the relationship of RNA and protein levels is 'controversial'; that's not really scientific. Indeed, this is the case either or both because there is too little data, data are wrong, or the right type of data have not been collected. I think this manuscript makes a great case for the last of these. I.e., prior work (arguably) has been too crude to ferret out or isolate different factors responsible. Here, having the gene the same but changing the 5' or 3' UTR is a much better controlled experiment to sort out these effects. I know that wasn't the authors' intention – they wanted to discover these effects – but serendipity has given them a great controlled experiment as well. My guess is that this is why they're able to get so much more information.*

We agree that prior work, which has largely focused on translation at the gene level, has not collected the right type of data and/or performed the right type of analyses to uncover isoform-specific translational control. We have removed the statement regarding “controversy” and reframed the Introduction to focus on limitations of prior datasets as advised.

*2) Abstract: A suggestion (strongly) not to use the term rheostat. Analogies can be helpful, distracting, or misleading. Here at a minimum the rheostat is not a physical analogy. Presumably the effects arise from molecular interactions – and the results lead to the neat model that 3'UTRs are cell type specific because different cells have different proteins expressed (providing additional evidence in support of the 3' UTR as hosting a complex panoply of proteins; the constancy of 5'UTR effects suggests that these arise from something common to all cells –perhaps the ribosome and translation initiation machinery.*

We have removed the term rheostat. The Abstract now reads: “These results expose the large dynamic range of transcript-isoform-specific translational control, and identify isoform-specific sequences that control protein output in human cells”.

*3) Introduction: The first paragraph is considerably under-documented. How well established are 'diverse translational control elements'? There are certainly a number of specific cases, but is their number and breadth clear? And the three examples given are maybe not the most clear or general. While one can't have a review here, an opening perspective that is both broader and contains important details would provide the scholarly context for this work.*

We have added additional citations to the first paragraph of the manuscript documenting classic examples of translational control elements, such as uORF-mediated control of GCN4, the binding of the iron regulatory protein to transcript 5’ leaders, and translational regulation by micro RNAs. We think that the number and breadth of translational control elements are not clear, which is part of the motivation for this study.

*4) Introduction: The authors point out that there is not a technique to look at isoform specific translation. They then take nearly a page to note how 'ribo-seq' cannot do this. They then reveal that sucrose (polysome) gradients, followed by deep sequencing, can do this. But of course polysome fractionation is a very old technique. And it is not until the Results (paragraph one, subheading “TrIP-seq measures transcript isoform-specific translation in human cells”) that we learn that Arava et al. did polysome fractions for all yeast genes – at that time using microarrays. And microarrays could be designed to look at isoforms, or looked at another way, the technical extension of this work from the existing technologies is a modest step, but not presented in the appropriate historical context. While this is common, it is not needed. The biology and importance of the question stands for itself. So the authors’ main advance is not a technique – but rather the identification of a (very) important question and unmet need, and the expert execution of addressing this question. (As a related aside, if one goes back to the original Ignolia et al. Science paper introducing ribo-seq, there the method has held – and been improved – and has been very important, but each of the scientific conclusions drawn was either made previously (by Arava et al.) or later shown to be incorrect. That paper still deserved the wide audience it got, as does this one, but let's work hard to get the intellectual and scholarly history right, whether or not it involves prior work that I was associated with.)*

We appreciate the view that this study addresses an important question that is interesting to a broad audience. Indeed, the motivation behind this study is not to develop a new technique but to determine the fundamental biology discussed here – how do transcript-specific sequences contribute to translational control? We apologize that this was not presented in the appropriate historical context and have edit the Introduction and Discussion to focus more on the scientific history and context for this work. We tried to strike a balance between selling the work and presenting the raw science and thank the reviewer for shifting us back to science.

5) Subsection “TrIP-seq measures transcript isoform-specific translation in human cells” and Figure 1:

*It is not clear that had Figure 1 not recapitulated the gradient order that it would have meant that there is something amiss in the experiment.*

We regret that the motivation behind this analysis was not clear. Since the sucrose gradient was fractionated manually, it is possible that technical errors could lead to tube swaps or mis-fractionated peaks. This analysis is meant to be a test of the relationship between the samples with regard to their origin in the gradient, and their clustering into the gradient order demonstrates that the fractions were faithfully collected. We have amended the sentence to read “Clustering of the samples recapitulates the gradient order (Figure 1), indicating the polysome profile was accurately fractionated”.

*While the clustering 'broke' between poly 4 and 5, this is not a statistically tested or rigorous result, and there are many possible origins. So the conclusion (from this observation at least) that the data can be considered in terms of low and high polysomes is not supported. One variable to consider is that short genes (which are highly abundant) will by necessity cluster in the low region. Thus, that statement “This suggests that in cells, transcript isoforms are predominantly poorly- or highly-ribosome associated, causing low polysomes to cluster away from high polysomes” is an over-interpretation of the data that are presented.*

We have performed statistical resampling of the cluster boundaries in Figure 1 to test the significance of the “break” in the clustering. We measured the Jaccard distance between clusterings of these data that were subsampled by bootstrap, subjected to jitter (addition of random noise to each point) or had random points replaced by noise using the R package fpc. The average Jaccard distance for each condition as well as the average Jaccard distance across the three tests is presented in Figure 1—figure supplement 1. A commonly used criterion to define robustly separated clusters is a Jaccard distance of 0.75, showing that on average the clustering in Figure 1 is robust.

Although short ORFs might cause the observed partitioning, this does not change the conclusion that “transcript isoforms are predominantly poorly – or highly –ribosome associated”; instead it suggests a biological basis for this conclusion. We have amended the sentence to read: “This suggests that in cells, transcript isoforms are predominantly poorly- or highly-ribosome associated, causing low polysomes to cluster away from high polysomes, which could be have numerous biological origins including highly abundant short ORFs”.

*Technical: Was the clustering done on all reads or on a per-gene basis?*

We have added a new section to the Methods to describe the inter-sample clustering “Clustering of TrIP-seq samples”.

*Technical: How is TPM (transcripts per million) determined? What controls for the validity of this measure are there?*

We apologize for leaving out a description of the unit transcripts per million (TPM). TPM is a unit of gene expression that has superior properties to the more-common reads-per-kilobase-per-million-mapped-reads (RPKM) by removing the dependence on sequencing library size. See e.g. PMID 22872506 “Measurement of mRNA abundance using RNA-seq data: RPKM measure is inconsistent among samples” by Wagner, Kin and Lynch. In our work, TPM is a transformation of RPKM as in Wagner et al:

TPMi=RPKMi 106∑all genesRPKMg

where RPKM was calculated using the Cufflinks suite. A description of this has been added to the Methods section. Using TPM improves comparability between studies but as it is linearly related to RPKM all the same controls for validity of RPKM, which is widely used, apply to TPM.

Technical: What controls for rapid freezing of translation upon cycloheximide addition were performed? The extent of reads near the 5' end of the ORF should be a measure of this and should be noted and documented.

We appreciate the concern for the acute stalling of translation by cycloheximide. Translational shutoff in mammalian cells by cycloheximide is known to occur rapidly (e.g. within 20 minutes and the fastest inhibitor tested in Han, K et al. Nature Methods 2014) and the concentration used in these experiments (~350 μM; 100 μg ml-1) is ~100 times higher than that required to inhibit protein synthesis in reticulocytes (Godchaux, W. et al., 1967). We cannot use reads near the 5’ end of the ORF as a control for this since reads in TrIP-seq are derived from RNA-seq and not ribosome protected fragments, but similar conditions acutely shut off translation in ribosome profiling protocols in HEK 293T cells where you can perform this analysis (Sidrauski et al., 2014). We have added this to the Methods section of the manuscript.

Figure 1: While it is true that noncoding are preferentially in low polysome fractions, they ARE in those fractions, and the difference between low and high is only 2x. Is there reason to be concerned by this? (The statement “Protein coding genes and isoforms are found across the polysome, but noncoding and lncRNA genes are predominantly in the low polysome fractions indicating that they are generally weakly ribosome associated or in other large macromolecular complexes (Figure 1 and Figure 1—figure supplement 1)” seems overstated given the 2x differential.)

Indeed, we were surprised by the number of noncoding genes present in high polysome fractions. We have analyzed noncoding genes in high polysomes but did not include it in this manuscript version because we are still determining what it means. For example, clustering of transcripts derived from annotated noncoding genes reveals a similar heatmap to that for coding genes, with some transcripts strongly polysome associated (Figure 8). Analysis of the gene types represented in high polysomes shows a two-fold enrichment for antisense transcripts (Figure 8). We validated numerous individual examples of polysome-associated antisense transcripts by comparing to ribosome profiling datasets from HEK 293T cells from the Cho or Ingolia labs, for example the gene pair MCM3AP and MCM3AP-AS1 (Figure 8). Therefore, one explanation for the abundance of noncoding genes in high polysome fractions is that a fraction of genes annotated as noncoding are translated. If this is interesting to the reviewers or editors we can include this analysis and more as a main text or supplemental figure.

Author response image 1.Polysome association of noncoding genes.(**A**) Clustering of noncoding transcript isoforms as in Figure 2. (**B**) The fraction of transcripts per noncoding gene type in high polysome clusters red and blue (gray) versus all annotated noncoding gene derived transcripts (black). (**C**) The gene *MCM3AP* and its antisense gene *MCM3AP-AS1* are polysome associated and exhibit ribosome profiling reads from two datasets, one of which is stranded (red reads: sense strand; blue reads: antisense strand).**DOI:**
http://dx.doi.org/10.7554/eLife.10921.023

*Figure 1: The description is inadequate – what are the points?*

We apologize that this was not clear and have revised the figure legend to read: “(F) Spearman’s correlation (RS) between gene (left) or isoform (right) read counts from ribosome profiling or TrIP-seq. See also Figure 1—figure supplement 1 and Methods for calculation of TrIP-seq polysome counts”. A description of the calculation of TrIP-seq polysome counts is in the Methods section now titled “Ribosome profiling comparison and TrIP-seq polysome count calculation”.

*6) “Agreement between the isoform-specific polysome profiles and RT-PCR conducted from polysome fractions verified the accuracy of the isoform-level quantifications (Figure 1)”. Figure 1 does not show even qualitative agreement between RT-PCR and TrIP-seq results. This gel and the one in the supplement should be quantitatively compared to the trip-seq data – quantitative PCR is probably needed here.*

See next response.

*7) “Weaker amplification of the longer isoform (EEF1B2-003) in high polysomes may be due to PCR bias towards smaller amplicons (Walsh et al., 1992), and replicate experiments show EEF1B2-003 present in high polysomes (Figure 1—figure supplement 1)”. The comment indicates less than robust data agreement, and a potential reason for this. That suggests that additional experiments (like shorter PCR reads) should be done to determine if the expected agreement is indeed observed.*

We appreciate the concern for rigorous validation of the deep sequencing data. It was not possible to make quantitative comparisons with these gels because it is not quantitative PCR as mentioned. Qualitatively, isoform 003 is primarily in low polysomes and generally decreases towards high polysomes, while isoforms 201 and 001 both qualitatively increase from low to high polysomes, both by RT-PCR and TrIP-seq, so there is qualitative agreement between the validation and sequencing data.

We have performed qRT-PCR with isoform-specific primers from polysome fractions and present these new data for *EEF1B2* and *SRSF5* in Figure 1—figure supplement 2. Again, qualitative agreement is observed but the relationship may be less quantitative than desired, perhaps due to the difficulty in accurately reconstructing transcript isoforms from short read sequencing data or different RNA normalizations between the two experiments. We believe the strongest validation of TrIP-seq is that it can be used to control the translation of reporters as predicted by the sequencing data as in Figure 5, and future work could use long read sequencing to directly measure transcript isoforms without the need for in silico reconstruction.

*8) Subheading “TrIP-seq measures transcript isoform-specific translation in human cells”. Where the trip-seq replicates completely independent biological replicates? If not, this conclusion cannot be drawn.*

We apologize for the confusion shared by both reviewers on this issue and have modified the Methods section to reflect that the two TrIP-seq datasets are independent biological replicates.

*9) “Therefore, TrIP-seq and ribosome profiling agree at the gene level but diverge at the isoform-level, likely due to difficulty in mapping the very short profiling reads at the isoform level, or the influence of untranslated regions, which are invisible to profiling”. As I understand it there are two models for the differences between full ribosome profiles and that for the isoforms – noise from fewer reads or 'real' effects. This isn't so clear from the wording here, and seems like it needs to be addressed here – since one is an artifact and the other is biology.*

We appreciate the request for clarification and have reworded this sentence to read: “However, at the isoform level the correlation is worse (R_S_ = 0.47), which is worse than the correlation between TrIP-seq replicates (R_S_ = 0.90; Figure 1—figure supplement 1), suggesting the discrepancy is not due to variability in isoform quantification in TrIP-seq but instead is likely due to known issues quantifying transcript isoforms in ribosome profiling (Ingolia 2014)”. The citation here is to a review by Ingolia wherein he specifically discusses the issues associated with quantifying transcript isoforms using ribosome profiling: “transcript variants with different 5ʹ and 3ʹ UTRs affect translation, and ribosome profiling cannot deconvolve these transcripts when multiple variants are present simultaneously”.

*10) “We then compared TrIP-seq data to copy number estimates from HeLa cells, and find that the two datasets correlate well (R_S_ = 0.55, Figure 1—figure supplement 1, (Kulak et al., 2014). “:R_S_ – 0.55 is viewed as good, but similar correlation coefficients are viewed as bad in other contexts. Maybe instead of saying correlate 'well' we need to better understand the factors that affect this correlation. Figure 1—figure supplement 1 shows a spread in Hela copy number (protein #) that looks much broader than the spread in trip-seq polysome counts. Visually I don't think a neutral reader would say that trip-seq data account well for the protein levels.*

We have now correlated TrIP-seq data against HEK 293T proteomics data and present the results in Figure 1—figure supplement 1, which also shows a positive correlation. We agree there is still considerable spread in this plot, but note that TrIP-seq is a better predictor of protein levels than RNA-seq (R_S, TrIPseq_ = 0.57, R_S, RNAseq_ = 0.45). We have revised this sentence to read: “We then compared TrIP-seq data to protein copy number estimates from HEK 293T cells, and find that the two datasets correlate”.

*Importantly, the rest of this paragraph is logically unclear – statements are made but the underlying evidence for those statements seems an opinion rather than a reasoned series of considerations. The final statement is strong: 'we conclude…' – but the basis for that conclusion does not appear strong or minimally is not clearly laid out. And these are the crucial data for that paper.*

We thank the reviewer for helping us to improve the clarity of this paragraph. We based our reasoning on a series of observations from our data and the literature: cryptic large complexes would systematically overestimate protein abundance, but in general HEK 293T protein abundances are lower than TrIP-seq abundances (Figure 1—figure supplement 1) and there is a high gene-level correlation with ribosome profiling (Figure 1), which is an accepted measurement of translation due to e.g. demonstrated ribosome-derived read length patterns (Ingolia 2014 Cell Reports). There are examples of systematic stalling of elongating ribosomes, which would also decouple ribosome occupancy and translation (citations in the manuscript), but we reasoned here based on the literature that this is a minor effect because of e.g. no detectable long-term ribosome stalls in mouse ES cells (Ingolia 2011). Previous work has related polysome profiles to translation (e.g. Arava et al.) and, in addition to the observations laid out here and in the Discussion, translating ribosome association is the most parsimonious explanation for why an mRNA is associated with a polysome peak. We have toned down the final statement of this section and reserve it for the Discussion, after testing the ability of these data to predictably control translation in Figure 5.

*11) Subheading “An atlas of human isoform-specific translation patterns”, Figure 2; clustering. Clustering is statistically ambiguous, ultimately, and these data are pretty 'flat' – i.e., there aren't many very strong features to cluster by. But maybe more centrally since clustering is a somewhat arbitrary (though often useful) transformation – e.g., there is no natural or set number of clusters for a dataset- it is not clear that one what's to be making conclusions – rather than just organizing data – from clusters. The authors conclude that it's 'crucial to fractionate the polysome profile.' I think that's strongly expected a priori, but I don't see how that was strongly tested or established from the cluster analysis – other than to note that there are some differences in profiles.*

Our goal with this analysis was to find a set of transcripts that are poorly and highly translated and then extract features of gene-linked isoforms that may be associated with differential translation. To do this, one must categorize transcripts somehow and we elected to use clustering. We provide evidence that the specific clusters referenced most in the paper (e.g. clusters 1, 2 and 6) are present in each biological replicate by measuring the Jaccard Index between biological replicate clusters and pooled replicate clusters (Figure 2—figure supplement 1).

The sentence referenced regarding: “fractionating the polysome profile” is “many clusters have similar total polysome abundance but different distributions, indicating that to obtain accurate measurements of isoform-specific translatability it is crucial to fractionate the polysome profile”. By this we mean that comparing total cytoplasmic RNA to total polysomal RNA as was done in Sterne-Weiler et al., Maslon et al., and other earlier studies is blind to the diverse transcript isoform specific translation patterns highlighted by the clustering analysis. Later we show that regions from isoforms which are associated with many or few polysomes are sufficient to control protein production by *Renilla* luciferase by 100-fold, so the strong test of this conclusion comes later.

*12) Paragraph two, subheading “An atlas of human isoform-specific translation patterns”. The conclusions in the paragraph seem too strong from the data. The authors above said that they don't know why ncRNAs are in the polysomes – which is reasonable. But here they make conclusions about decay and pioneering rounds – these arguments are likely not germane and do not seem to be supported by the data or logic but rather seem to be 'fits' to existing ideas and models.*

We have removed this paragraph and Figure 2.

*13) “Surprisingly, the poorly translated cluster 7, which has ~9,000 individual transcripts…”. It is not clear to me that seeing ~10K sequences with introns means that they are being translated. That's one model. The other, stated earlier in the manuscript, is that they are in other large complexes.*

We were curious about this point as well, and inspected a variety of retained intron transcripts in other ribosome profiling datasets to see if ribosome protected fragments could be found on these transcripts. In numerous examples we find low levels of ribosome profiling fragments derived from retained intron regions, confirming that at least some of the ~10k retained intron transcripts are indeed engaged with ribosomes (Figure 9). Computing the FLOSS score of these retained intron derived fragments shows they are indistinguishable from coding sequence fragments (Figure 9). We are happy to provide further examples of retained intron transcripts with ribosome protected fragments and corresponding analysis as a main text or supplemental figure if the reviewers or editors are interested.

Author response image 2.Retained intron transcripts found in polysomes are also observed by ribosome profiling.(**A**) An example of a retained intron transcript on polysomes where the retained intron region is detected through the eight+ ribosome fraction. (bottom) Ribosome profiling reads are detected in the retained intron region in two separate 293T ribosome profiling datasets. (**B**) Shown are fragment length distributions for ribosome profiling reads derived from either retained intro regions (red) or coding sequence regions (black). The FLOSS score (39) between retained intron and CDS fragment length distributions is 0.023, indicating near perfect agreement. The FLOSS score is the integrated point wise difference between the two distributions. Ribosome profiling data from the Yoon-Jae Cho lab and from [79].**DOI:**
http://dx.doi.org/10.7554/eLife.10921.024

*14) “In sum, clustering of isoform abundance distributions across polysomes immediately provides insight into the diverse patterns of ribosome association by individual isoforms and transcript types in cells, and reveals intron-rich clusters of transcripts associated with polysomes that escape nuclear detention”. For similar reasons, I don't see how this conclusion is reached, relative to other possibilities.*

We apologize that the reasoning behind this statement was not clear. The retained intron transcripts observed in these data escape the nucleus since the input material to the sucrose gradient was from the cytoplasmic fraction of cells. If desired, we can include explicit demonstrations of the difference between the abundance of retained intron regions between the nuclear fraction and polysome distribution of retained intron isoforms, as can be seen on the left and right sides of Figure 8, “nuc” row, which shows sequencing data from the nuclear fraction of the same cells used for the TrIP-seq experiment.

*15) Subheading “3′ UTRs and introns drive isoform-specific polysome association”. Here the authors are using the clusters to explore and identify possible trends. This seems like a powerful exploratory use of the clusters.*

We appreciate the comment; this analysis is the reason the data were clustered in Figure 2 and was the major motivation of the entire project!

*16) “We find that longer coding sequences and highly abundant versions of gene-linked isoforms are biased towards high polysomes (Figure 3 and Figure 3—figure supplement 1)”. This can be because short genes can't hold as many ribosomes – or they reach the end sooner and thus fall off sooner.*

We agree, and have added wording to this effect.

*17) Figure 3, subheading “3′ UTRs and introns drive isoform-specific polysome association”. A number of interesting points are raised and discussed. The bottom line at the end of this section seems vaguer than one would like – e.g., what is really established vs what might be and what might have other explanations. More basically, is the best way to analyze these data from pre-clustered groups or, since the top and bottom fractions are compared, to use a numerical ratio of those – so that one analyzes a continuous rather than discretized variable (as this distorts error).*

We chose to use the clustered data rather than numerical comparisons across all transcripts because 1) the clustering allowed us to extract isoforms that were especially lowly or highly polysome associated (which should then have the largest differences), and 2) difficulty in converting the TrIP-seq polysome to a precise number of ribosomes associated with each transcript since the last polysome peak is degenerate. There are multiple ways to analyze these data and other types of analysis would also be interesting in the future.

*18) Subheading “Predicting translation changes during preimplantation human development”. What does 'reprocessed' mean? It's not clear from what's stated what this dataset is? Presumably it is not polysome data – so presumably it's seeing whether gene isoforms change in abundance (lit data) that have isoform specific differences (this data).*

We apologize for the vague use of “reprocessed”. We meant that since the original data were quantified using a different set of analysis software and onto a different set of reference transcripts the two datasets could not be directly compared as is. The latter interpretation is correct; the purpose of this figure was to demonstrate that further information can be distilled from existing, precious sequencing data (like that collected from human embryos) when interpreted given the unique insight into the translatability of each transcript isoform that TrIP-seq provides. We have added a reference to the Methods section on this line and the full reprocessing is described in the Methods section “Reprocessing of embryo sequencing data”.

*19) Paragraph two, subheading “Predicting translation changes during preimplantation human development”. Seems like general conclusion about use of trip-seq from one example – and even that example doesn't seem so striking, and given that there are 1000's of data, the chance of a small number having seemingly statistically significant effects seems high. This might be considered as some form of multiple hypothesis testing.*

We have included three additional examples of transcript isoforms that change across embryonic development and are also differentially associated with polysomes in the new Figure 4—figure supplement 2.

*20) It seems like the best independent test of the data and models is predicting new effects, and that's what this section does. The key statement seems to be “Not only do these data provide strong evidence that the output of heterologous reporters can be predictably tuned by untranslated regions, it additionally validates that the polysome profiles observed are connected to translational output”, stating that the results have predictive value. But the comparison is complex – i.e., showing polysome profiles. There are standard ways to convert polysome occupancy into protein production rate. While imperfect, that's the basis for earlier stuff in this and other work. So what seems clearest and least biased would be to make quantitative predictions for several – preferably 10-20, variants, and compare the output to those quantititave prior predictions. Figure 6 (below) suggests that this should be doable for 5' leaders but not 3' UTRs.*

We have performed quantitative comparisons between the predicted protein output and observed luciferase levels and present these in Figure 5. There is a positive relationship between the average number of ribosomes in TrIP-seq measurements (Figure 5) and the luciferase fold change across all reporters (Figure 5). It is challenging to convert TrIP-seq abundances to quantitative protein synthesis rates due to the poor gradient resolution of mRNAs associated with many ribosomes in the poly8+ peak. This is one possible source for the variability in 5E. Testing additional variants as well as the mechanistic foundations of the differential translation observed is the subject of other current work and we agree it will be fruitful to conduct this comparison.

*21) Discusssion. Where does the value of two orders of magnitude come from?*

We measured luciferase production modulated by untranslated regions derived from different gene-linked isoforms and found one example where two isoforms affect protein production by ~100-fold, and three other examples that affect protein production by ~10-fold. We apologize that this was not clear and have added a reference to Figure 5 with this statement.

*While no large translational regulation was found in prior work with 3000 highly expressed genes, are there not examples of translational regulation of genes in humans? Or in related species with conserved genes?*

We have removed this statement and added references regarding translational control at the gene level to the Introduction.

*Paragraph two, Discussion. As noted above, the statement that trip-seq provides a 'faithful measure' may not be well supported by the analyses presented.*

*It seems that by far the strongest support for the manuscripts conclusions is the 'engineering', though that can be improved as noted above.*

We agree that some of the strongest support that the TrIP-seq data is a measure of translation is derived from the reporter experiments. Given this, the discussion earlier in point 10, the discussion presented in paragraph three, Discussion, and the fact that the most parsimonious explanation for a mRNA being in a polysome profile is that it is associated with a ribosome, we stand by the statement that TrIP-seq by-and-large is a faithful measure of isoform-specific translation in cells. There are certainly exceptions to this conclusion; the Discussion is speaking to the majority of the signal in TrIP-seq.

*22) An interesting test would be to demonstrate/test isoform-specific effects of a miRNA.*

This would be a fascinating experiment and we will consider the best way to perform it. This is one of many follow-up experiments that could be performed given the observations here regarding isoform-specific translation, like altering the levels of a miRNA, sequence-specific RNA binding protein, DEAD-box protein, or ribosomal protein. Experiments like this are uniquely possible with measurements of translation with transcript isoform resolution provided by TrIP-seq, and we are excited to pursue experiments in this vein.